# Efficiency stagnation in global steel production urges joint supply- and demand-side mitigation efforts

Peng Wang[1,2], Morten Ryberg [3✉], Yi Yang[1,4,5], Kuishuang Feng [6,7], Sami Kara [2✉], Michael Hauschild[3] & Wei-Qiang Chen [1,8✉]

Steel production is a difficult-to-mitigate sector that challenges climate mitigation commitments. Efforts for future decarbonization can benefit from understanding its progress to date. Here we report on greenhouse gas emissions from global steel production over the past century (1900-2015) by combining material flow analysis and life cycle assessment. We find that ~45 Gt steel was produced in this period leading to emissions of ~147 Gt $CO_2$-eq. Significant improvement in process efficiency (~67%) was achieved, but was offset by a 44-fold increase in annual steel production, resulting in a 17-fold net increase in annual emissions. Despite some regional technical improvements, the industry's decarbonization progress at the global scale has largely stagnated since 1995 mainly due to expanded production in emerging countries with high carbon intensity. Our analysis of future scenarios indicates that the expected demand expansion in these countries may jeopardize steel industry's prospects for following 1.5 °C emission reduction pathways. To achieve the Paris climate goals, there is an urgent need for rapid implementation of joint supply- and demand-side mitigation measures around the world in consideration of regional conditions.

[1] Key Lab of Urban Environment and Health, Institute of Urban Environment, Chinese Academy of Sciences, Xiamen, China. [2] Sustainability in Manufacturing and Life Cycle Engineering Research Group, School of Mechanical and Manufacturing Engineering, The University of New South Wales, Sydney, Australia. [3] Quantitative Sustainability Assessment Group, Sustainability Division, Department of Technology, Management and Economics, Technical University of Denmark, Kgs, Lyngby, Denmark. [4] Key Laboratory of the Three Gorges Reservoir Region's Eco-Environment, Ministry of Education, Chongqing University, Chongqing, China. [5] Environmental Studies Program, Dartmouth College, Hanover, NH, USA. [6] Institute of Blue and Green Development, Shandong University, Weihai, China. [7] Department of Geographical Sciences, University of Maryland, College Park, MD, USA. [8] University of Chinese Academy of Sciences, Beijing, China. ✉email: moryb@dtu.dk; s.kara@unsw.edu.au; wqchen@iue.ac.cn

Steel is the most used metal in our modern world, but its production is highly energy- and carbon- intensive. To achieve a climate-safe future as required by the Paris Agreement, there is a need for reaching net-zero emissions by around 2050 and net negative emissions thereafter[1,2] for every sector including the steel industry[3,4]. However, steel production, together with other energy-intensive industries such as cement and petrochemicals, is considered a difficult-to-mitigate sector[5–7]. Their decarbonisation remains extremely challenging on the following grounds: First, their global demands are projected to increase to support a growing and increasingly affluent population[8,9]. Second, some carbon-based resource is essential for high-temperature heat and steelmaking and cannot be easily replaced[7,10,11]. Third, the long-lived facilities in their production may further hinder the required mitigation progress due to the carbon lock-in effect[12,13]. Thus, compared to transportation and energy sectors, the corresponding innovation, progress and understanding related to the decarbonisation of global steel industry are generally lagging behind[14–16].

Strategies for decarbonising the steel industry have primarily focused on production efficiency improvement, including energy efficiency measures[10,17,18], production technologies innovation[19,20] and fuel switching[21–23]. However, the effectiveness of such production-based strategies in terms of carbon reduction has recently been questioned[3,14,24,25]. This calls for attention to gauge the entire progress that the global steel industry has made on GHG mitigation. Most of previous investigations have been limited to specific production technologies[24,26,27] where the interplay between material flows and supply-side technical efficiency was widely overlooked. Such lack of understanding could prohibit the development of strategies that are more effective for steel industry toward future GHG emission mitigation.

Here, we integrated dynamic material flow analysis (MFA) with life cycle assessment (LCA) to estimate annual production, efficiency and GHG emissions of global steel production based on 19 dominant processes during 1900–2015. By examining the interplay between material flows and GHG emissions, we found an exponential increase in steel production volume (ca. 3.4%/year) and associated GHG emissions (ca. 2.5%/year) over the past 115 years, despite a concomitant reduction in the carbon emissions intensity by ~67% achieved through technical innovation and efficiency improvement. We then performed a decomposition analysis to reveal the contribution of efficiency improvement and production outputs to these emission changes, the results of which highlighted the inadequacy of process efficiency alone in achieving absolute emissions reduction. We found that the GHG intensity of the global steel industry had stagnated in the past 15–20 years before 2015. By region specific investigation, we found that there were improvements in technology efficiencies during the past few decades but these were offset by the concomitant growth of steel production with low process efficiency (especially in China and India). This stagnation indicates the urgency of the joint implementation of process efficiency and demand-side measures to reduce GHG emissions and achieve climate targets.

## Results and discussion

### GHG emissions weigh three times more than the steel produced.
We estimate that global steel production emitted a total of ~147 billion tonnes (Gt) $CO_2$-eq from 1900 to 2015, accounting for ~9% of global GHG emissions during this period (see Fig. 1 for each process, with additional details in Section S1.2). The iron-making stage contributed the most (around 50%) to the total emissions, caused mainly by the use of carbon as a fuel and as a reductant in the blast furnace (i.e., 58 Gt)[27]. The steelmaking stage

(excluding iron foundry) emitted 33 Gt of $CO_2$-eq totally, of which around half pertained to the open-hearth furnace (Fig. 1a). Despite a much lower carbon intensity (Fig. 1b), the steel finishing stage emitted 27 Gt $CO_2$-eq mainly due to the vast production flows (Fig. 2c). Over the studied period, the total GHG emissions from mineral treatment were 18.7 Gt $CO_2$-eq and this number is expected to increase in the future as a result of decreasing ore grade. The entire production system can be divided into two major production routes (see Fig. S1.2), i.e., the primary route with the ore-blast furnace system and the secondary route with scrap-electric arc furnace system. They differ substantially in terms of efficiency, resource use, emissions, and production volumes[28–30]. The secondary production route was around one-eighth as carbon-intensive as the primary route[31], and accounted for ~5% of total annual GHG emissions in 2015 as shown in Fig. 2d, e. Historically, it is estimated that the primary production route emitted 132 Gt $CO_2$-eq over the studied period, accounting for over 90% of the total GHG emissions from steel production.

Figure 3a presents the cumulative flows and stocks of steel along its material cycle from 1900 to 2015. Our analysis shows that the steel industry consumed ~46 Gt iron ore and ~31 Gt home, new and old scrap to produce ~45 Gt steel products during this period, which, in terms of weight, is around one-third of the total steel production-related GHG emissions (147 Gt $CO_2$-eq). At present, over half of those steel products remain as societal in-use stocks (i.e., ~25 Gt) with the largest share stored in buildings (~16 Gt) which were mainly constructed in the past 20 years (driven by large emerging economies, including China and India[32]). In general, those societal in-use stocks are quite young with ~83% of global steel in-use stocks being built after 1990 (Fig. 3a). Given that the average lifetime of steel products is ~70 years[9], a rapid increase in old scrap generation can be foreseen in these countries over the next 30–50 years. Indeed, the past few decades have already witnessed a significant increase in old scrap generation from ~45 Mt/year in 1950 to ~427 Mt/year in 2015 (Fig. 2g), concomitant with a remarkable improvement in steel recycling rate (now remaining at around 70%) (Fig. 2h). However, such an increase may not lead to a net decrease in steel production-related GHG emissions, as this will depend on the future increase in total steel demand and whether it can be, to a large extent, satisfied by secondary steel production[33]. Unfortunately, our analysis shows that these improvements have not kept up with the fast growth in steel consumption in recent decades as in-use stocks in emerging economies like China were too young (average age: 8.6 years in Fig. 3a) to generate enough old scrap, forcing regional steel production to rely heavily on iron ores[34]. This led to a notable decrease in the share of secondary production relative to primary production from 30% in 1995 to 21% in 2015 (Fig. 2i), contributing partly to the increase of the sector's total GHG emissions.

### Efficiency improvement offset by demand growth.
Given that energy constitutes 20–40% of total steel production costs[35], steel producers have a strong incentive to improve their energy efficiency. Our analysis shows that the steel industry has reduced GHG emissions intensity by ~67% since 1900 (Fig. 4a), and the steepest decrease occurred before 1940 due to energy efficiency improvement in the blast furnace[27]. After the 1940s, the largest drop in emissions intensity occurred between 1970 and 1995 (from ~4.5 to 2.6 t $CO_2$-eq/t steel), due to the improvement in energy efficiency through technological advances, such as the use of pelletizing in lieu of sintering for ore preparation and increased use of BOF instead of open-hearth furnace. Moreover, the continued decarbonisation of the electricity grid since the energy crisis in the 1970s has also contributed substantially to lowering the GHG intensity of steel production (Table S2.5–S2.6).

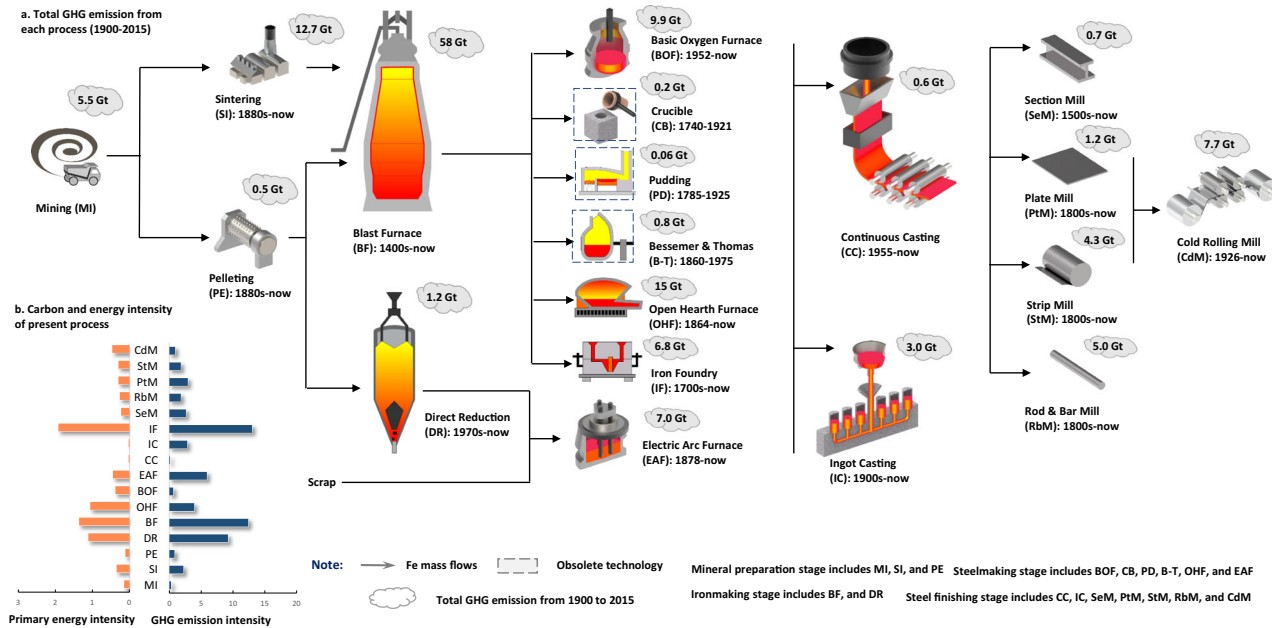

**Fig. 1 Steel production technologies and their total GHG emissions from 1900 to 2015. a** The connection of 19 dominating processes with their representative production technologies, application period and total carbon emissions from 1900 to 2015. **b** Energy intensity and carbon intensity level for each process (for details see Supplementary Information 1 Section S2.2). Abbreviations for steel production flows are: MI mining, SI sintering, PE pelleting, BF blast furnace, DR direct reduction, BOF basic oxygen furnace, CB crucible, PD puddling, B-T Bessemer & Thomas, OHF open-hearth furnace, IF iron foundry, EAF electric arc furnace, CC continuous casting, IC ingot casting, SeM section mill, PtM Plate Mill, StM strip mill, RbM rod bar mill, CdM Cold rolling mill.

Despite the decreased GHG intensity, the total GHG emissions associated with steel production increased substantially (2.5% per year, Fig. 2d) from ~0.22 Gt $CO_2$-eq in 1900 to ~3.7 Gt $CO_2$-eq per year in 2015, because of the exponential growth in the production of iron and steel products (on average 3.4% per year, Fig. 2a–c). The emissions increase indicates that the global steel industry has lost the "race between increasing consumption and efficiency gains"[36]. The demand growth was closely linked to national and global progress in urbanisation and industrialisation, as steel is the most fundamental metal for modern society. Consequently, the steepest production rise occurred in recent decades due to the rapid demand growth in emerging economies like China and India. (Fig. 3b). Correspondingly, ~45% of the total GHG emissions from the global steel industry occurred after 1990. Previously dominant technologies like Puddling, Crucible, Bessemer, Open-hearth furnace and Ingot casting were thus within a few decades almost entirely replaced by alternative technologies like basic oxygen furnace and continuous casting (Fig. 2b).

**GHG intensity stagnating since 1995.** Notably, the GHG intensity of steel production has stagnated at ~2.5 t $CO_2$-eq/t steel since 1995 (Fig. 4a, c, e). Our results show that the process efficiency of global steel production appeared to have stalled and levelled off (see Fig. S2.6 for detailed GHG intensity trend at the technology level) after 1995, which is consistent with plant-level investigations of the largest steel producers[37]. When combining technologies into the two steel production routes, the GHG intensities of both the primary and secondary production route have stagnated and remained at ~2.8 t $CO_2$-eq/t steel and ~1.0 t $CO_2$-eq/t steel (Fig. 4c, e), respectively. At the same time, the share of steel production in global GHG emissions increased from 5.2% in 1995 to 7.7% in 2015 (Section S4.1).

The historical trend suggests that efficiency improvement in steel production is important but insufficient to achieve net emissions reduction: 3.2-fold reduction in GHG intensity was

accompanied by 17-fold increase in total emissions due to the increase of steel production. We decomposed the impact of production flows (volume effect) and GHG intensity (efficiency effect) on total emissions with the Logarithmic Mean Divisia Index (LMDI) decomposition analysis (see Methods section and Section S4.2). During the studied period, the volume effect increased total emissions by ~23 Gt $CO_2$-eq, while the efficiency improvement reduced them by 6.9 Gt. Our results show that the efficiency improvement failed to outpace the volume increase in most of the 5-year period (Fig. 4b), and the volume factor dominated the emissions with a correlation coefficient of 0.97 (Fig. S4.3). However, a few exceptions occurred in the mid-1970s and 1980s (i.e., energy crisis period in Fig. 2f) when total emissions declined at an annual rate of 1.4%. These periods provide inspiration for future mitigation strategies. With spiked energy prices, the production costs for steel producers increased drastically, stimulating the quest for energy efficiency improvement to maintain total costs[38]. Thus, process efficiency was significantly improved in these periods (Fig. 4a, c, e), mainly in primary production through the adoption of emerging, more efficient technologies as previously mentioned at the steelmaking stage and through the promotion of energy-saving practices in Blast Furnace. In parallel, the energy crises also lowered total steel demand, which slowed the growth of steel production. In short, total GHG emissions were reduced as a result of changes on both production and consumption sides.

**Regional technical efforts are insufficient for global improvement.** During 1995–2015, steel industries in different regions have introduced various commitments, measures and innovations to improve process efficiency. For instance, Europe, as one of the most efficient steel-producing regions, has made various efforts (e.g., ULCOS programme started in 2004[39]) to reduce its carbon intensity by nearly 50%[40,41]. Efforts have also been made in Japan (e.g., the COURSE50 project started in 2008[42]), the USA (30%

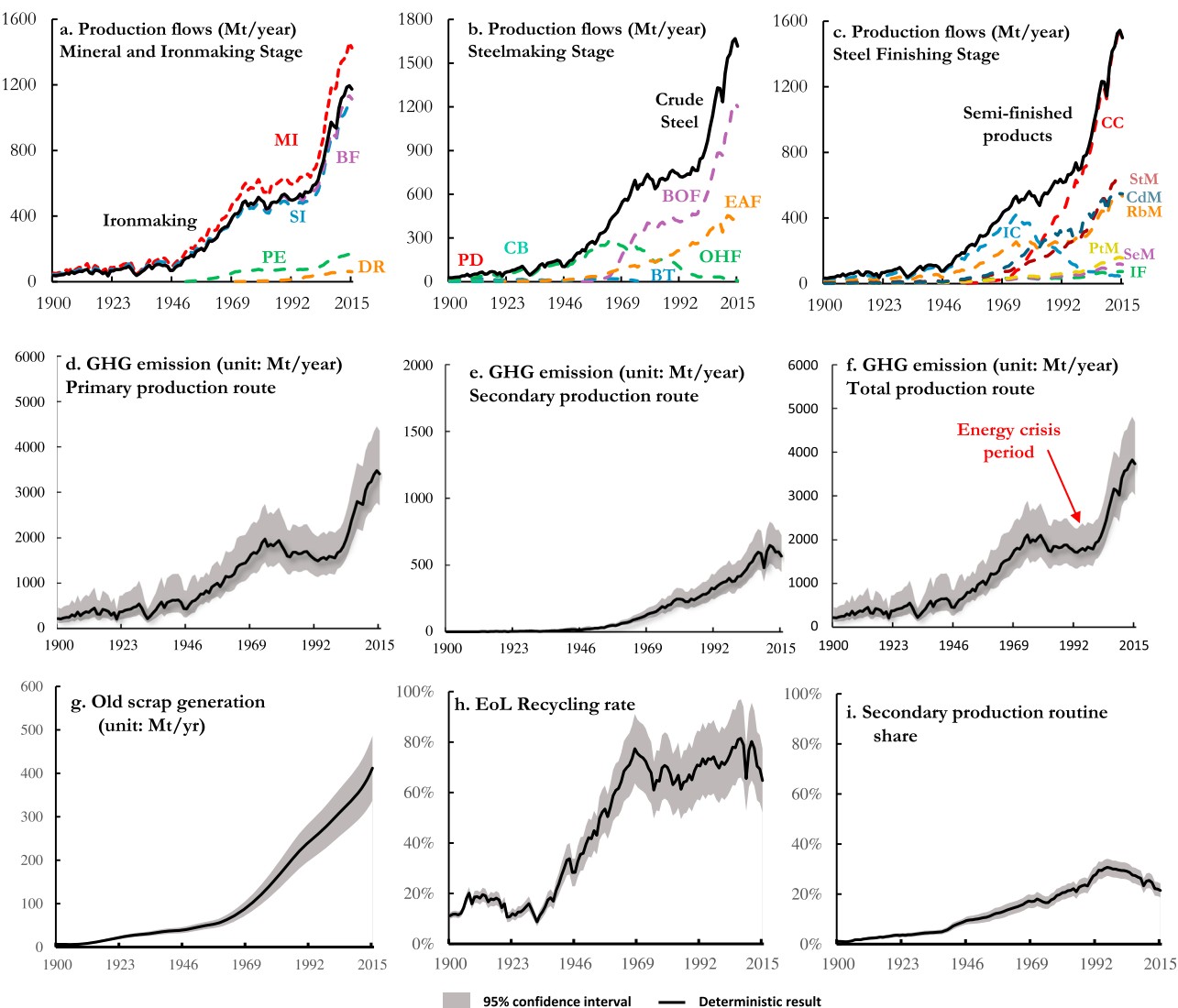

**Fig. 2 The steel production technologies and their annual production and GHG emissions from 1900 to 2015. a–c** Annual production flows from each production technology, derived from dynamic material flow analysis (details see Supplementary Information 1 Section S3.1, Unit: Million tons/year). **d–f** Annual greenhouse gas (GHG) emissions from primary, secondary and total production routes. The GHG emissions represent the sum of scope 1–3 emissions taking a life cycle assessment approach with scope 1 covering the direct emissions from the production site, Scope 2 including the indirect emissions from the energy used for the production and Scope 3 including indirect emissions associated with other inputs used in the steel production. **g** The historical trend of old scrap generation. **h** The estimated End-of-life (EoL) steel scrap recycling rate, calculated year by year from our dynamic material flow analysis. **i** The relative contribution of the secondary production route to total steel production. Data for **d–i** are presented as the deterministic results and the shaded areas indicate the 95% confidence interval of the estimates. Abbreviations for steel production flows are: MI mining, SI sintering, PE pelleting, BF blast furnace, DR direct reduction, BOF blast oxygen furnace, CB crucible, PD puddling, BT Bessemer & Thomas, OHF open-hearth furnace, IF iron foundry, EAF electric arc furnace, CC continuous casting, IC ingot casting, SeM section mill, PtM plate mill, StM strip mill, RbM rod bar mill, CdM cold rolling mill.

energy intensity reduction since 1990[43]) and other nations. With the help of continuous technological advancements and early retirement of redundant inefficient facilities, China has also made progress in improving its steel production efficiency (by ~30%[44]) during the past few decades. Moreover, the state-of-the-art ironmaking technology blast furnace is approaching the practical minimum energy requirement[31]. Nevertheless, those regional technical improvements are found to be insufficient to reduce the overall GHG intensity (let alone the total GHG emission) of steel production at the global level (as indicated in Fig. 3b, c) due to the structural changes in regional production flows.

By categorising those regions into different groups based on their emissions intensities (i.e., Tier 1, 2 and 3 in Fig. 3b), we find

an 8-fold expansion of crude steel production flows from the most carbon-intensive regions (i.e., Tier 3; Fig. 3c), rising from 129 Mt/year in 1995 to 914 Mt/year in 2015. By contrast, the operating production capacity from low (Tier 1) and medium (Tier 2) carbon-intensive regions has shrunk in the same period (Fig. 3b). Indeed, the share of global steel production in Tier 1 and 2 regions decreased from 83% to 43% during the studied period. The change in production flow structure has offset the mentioned regional technical advances. Notably, the poor carbon performance of Tier 3 regions was not simply attributed to their technical backwardness[45], but also closely linked to the distribution between primary and secondary production routes. Emerging economies have been fuelled by the fast expansion of steel

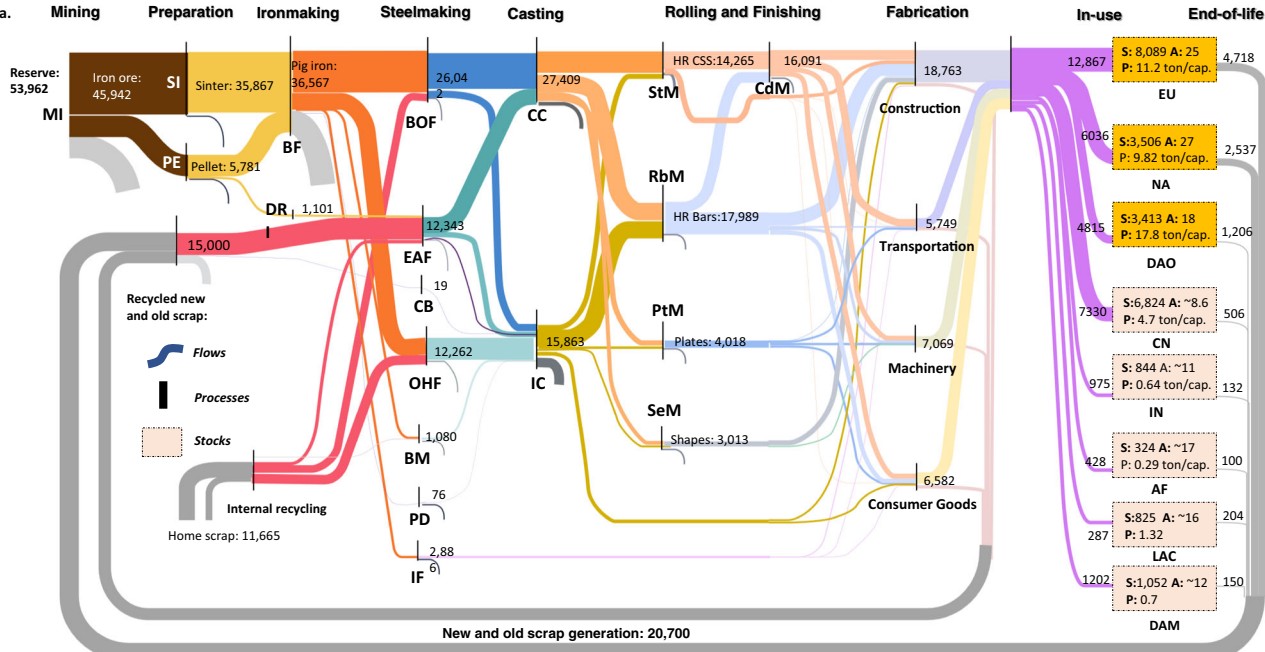

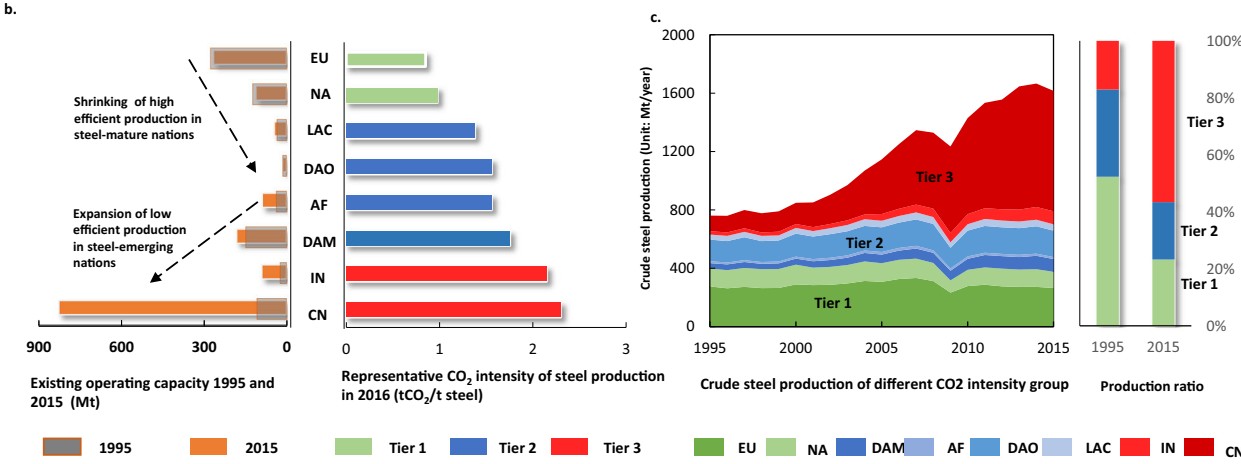

**Fig. 3 Historical steel flows and stocks along with its life cycle, and the recent trend of regional flows since 1995. a** The global historical steel cycle in a Sankey diagram where the numbers represent the accumulated annual flows over the past 115 years. **b** The change in operating production capacity for each studied region between 1995 and 2015, which is assumed to equal annual crude steel production based on data from World Steel Association yearbook[73]. The $CO_2$ intensities of these regions are obtained based on the benchmark investigation of Global Efficiency Intelligence[45] where only a limited number of nations is collected for the comparison of the situation[45] in the year 2016. The boundary for this calculation differs from ours, so we only categorised these nations into three efficiency tiers (i.e. EU and NA in tier 1, LAC, DAO, AF and DAM in tier 2 and IN and CN in tier 3) for indicative analysis. **c** The annual trend of steel production in the studied regions, which demonstrates that the historical growth in production flow was driven by the emerging nations (China, India, etc.) with poor efficiency performance (described in Section S4.3) as shown in their share change between 1995 and 2015. The abbreviations for regions are: EU Europe, NA North America, DAO Developed Asia and Oceania, AF Africa, DAM Developing Asia and Middle East, LAC Latin America and the Caribbean, IN India, CN China. Abbreviations for steel production flows are: MI mining, SI sintering, PE pelleting, BF blast furnace, DR direct reduction, BOF blast oxygen furnace, CB crucible, PD puddling, BT Bessemer & Thomas, OHF open-hearth furnace, IF iron foundry, EAF electric arc furnace, CC continuous casting, IC ingot casting, SeM section mill, PtM plate mill, StM strip mill, RbM rod bar mill, CdM cold rolling mill.

flows (production and consumption) and limited scrap availability. Thus, the production was dominated by the primary production route, which resulted in high GHG intensity for the entire steel industry (see the comparison between the USA and China[46,47]). Accordingly, our results highlight the necessity of coordinating the implementation of low-carbon technologies while also considering the regional structural changes in steel flows (particularly for emerging economies) to achieve deep decarbonisation of global steel production.

**Achievement of 1.5 °C climate target is jeopardized**. Future steel flow projections indicate a continuation of the observed historical pathway: steel demand in emerging economies continuing to grow and the ratio of scrap-based production flow remaining at 20–30% till 2035. As a consequence, the global GHG emissions intensity will unlikely to decrease in the near future. The recent World Steel Association's statistics[48] has also confirmed that emissions intensity had been stagnant from 2015 to 2019. In connection to the 1.5 °C climate target, this means that 37% of the GHG emissions budget

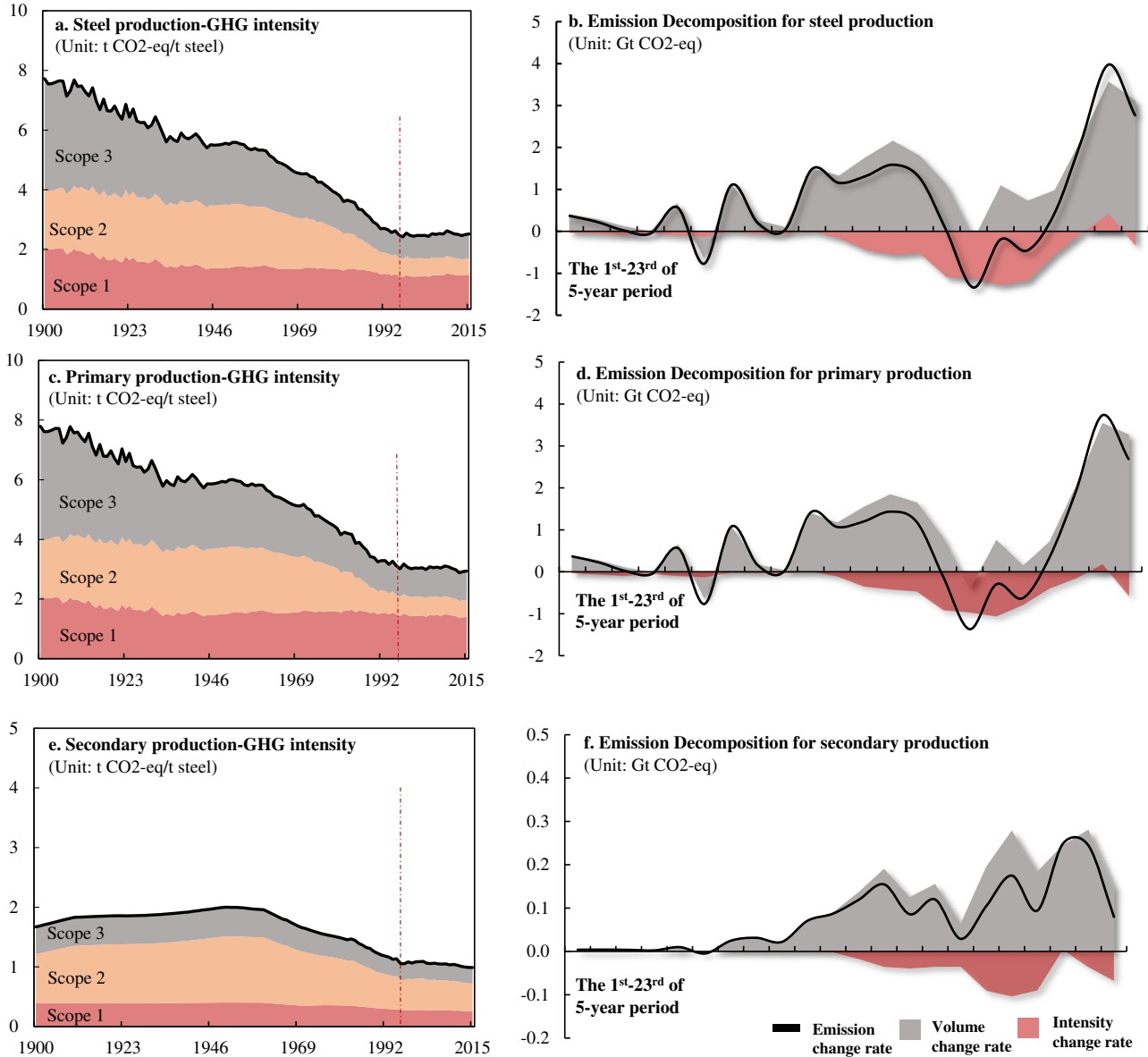

**Fig. 4 Improvement in the steel process efficiency and its linkages with production volume and GHG emission. a, c, e** The historical GHG intensity evolution in scopes 1, 2 and 3 for the total, primary and secondary production routes respectively. The red dot line highlights the start of a staggering period (see Supplementary Information 1 Section S4.3). We further decomposed the factors of volume and intensity from total emission impact (see Supplementary Information 1 Section S4.2). **b, d, f** The changes in emissions, volume and intensity during the 1st to 23rd 5-year period for total, primary and secondary production routes, respectively.

for steel production until 2050 has already been exhausted[49]. Furthermore, if the stagnation trend continues (as assumed in our BAU scenario; S1 in Fig. 5b), the entire carbon budget for steel production until 2050 could be fully exhausted by around 2035 to meet the growing steel demand. Meanwhile, the remaining carbon budget will also be exhausted before 2040 despite the implementation of either low-carbon technologies (S2 in Fig. 5b) or material efficiency (S3 in Fig. 5b) as indicated in IEA SDS (sustainable development scenario) trend[31]. Accordingly, we revisited previous GHG emission projections and found that meeting the 1.5 °C climate target[49] is unlikely unless we achieve a radical and immediate intensity reduction with an average rate of 0.85 t $CO_2$-eq/t steel per decade to become fully carbon-neutral by 2047 (S4 in Fig. 5b) or an additional 34% reduction in steel demand[31] (S5 in Fig. 5b). It will require a rapid innovation and implementation of low-carbon technologies to achieve the necessary reduction in GHG intensity. We summarised

37 types of breakthrough technologies in Section S4.5 and grouped them into seven categories: (a) Hydrogen-based options, (b) Electrolysis-based options, (c) CCUS with direct/smelting reduction, (d) Biomass-based options, (e) Blast furnace-improvement, (f) Carbon-free EAF and (g) Low-carbon rolling technologies. In combination, the breakthrough technologies have the potential to reduce GHG emissions at the required rate to become carbon-neutral by 2047. However, the rate of technology development and implementation is critical. To realise the 1.5 °C climate target, breakthrough technologies must be developed to a level that is fully operational and be implemented at global scale. Based on the review of the 37 breakthrough technologies, the rate of development appears too slow as most technologies are planned to be available in 10–25 years and with only limited implementation at that time. Aside from supply-side technology measures, various demand-side mitigation measures targeting material flows and technically

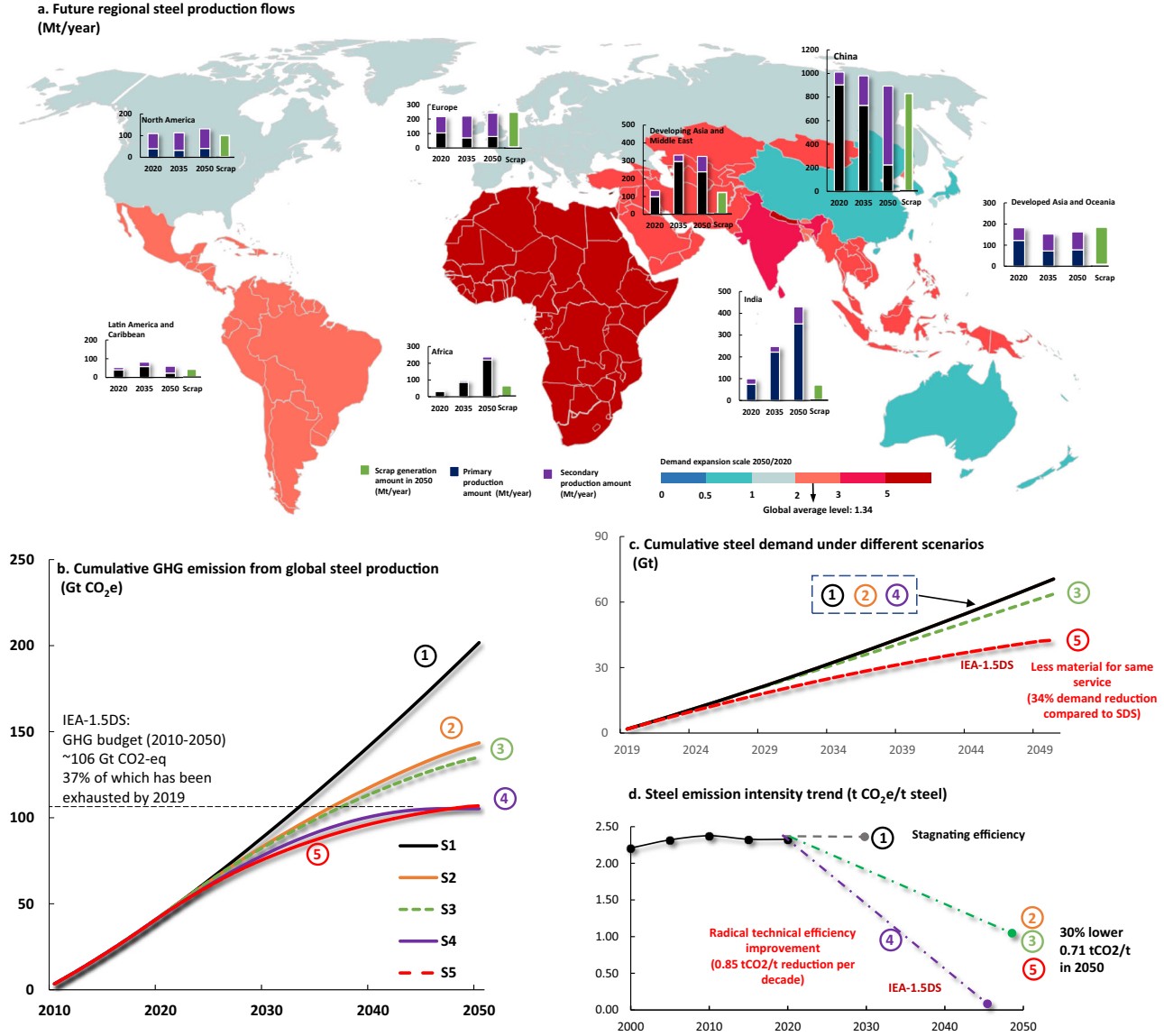

**Fig. 5 Feasibility of material and technical efficiency improvement scenarios to avoid the exhaustion of steel-related 1.5DS carbon budgets.** Scenarios are intended to represent explorative estimations rather than actual projections, and their settings are given in Supplementary Information 1 Section S4.4. **a** The regional growth of steel production flows from primary and secondary production routes (obtained from IEA Stated Policy Scenario (SPS) from its latest steel report[31]) and scrap generation (our calculation of scraps is based on the method from Pauliuk et al.[9]). We also adopt scenario analysis of required efficiency improvement trends under the 1.5DS carbon budget based on the projections of global steel production flows. **b**, **c**, **d** Six potential scenarios are generated for global steel production flows, i.e., Scenario 1(S1) BAU with ongoing stagnating efficiency trend with IEA SPS demand trend, Scenario 2(S2) with technical efficiency improvement under IEA's sustainable development scenario with 30% intensity reduction by 2050, and Scenario 3 (S3) with material efficiency improvement under IEA's sustainable development scenario with 12% reduction of total steel demand. All scenarios 1–3 will exhaust the carbon budget before 2050 as shown in Fig. 5b. Thus, Scenario 4(S4) and Scenario 5(S5) test the potential need of radical technical efficiency improvement (i.e. 0.85 t $CO_2$-eq/t steel reduction per decade) and material efficiency improvement (i.e. 34% additional demand reduction compared to IEA Sustainable Development Scenario) as ways to meet the budget constraint.

assisted lifestyle change beyond the direct control of the industry have received attention[25,50,51].

Given the pressing carbon constraints on the steel industry, we argue the need for integration of supply-side and demand-side measures to meet the 1.5 °C climate target. Indeed, a combination of supply- and demand-side measures would entail less radical reduction measures on both sides compared to if either side would need to achieve the reductions alone. We will not venture into the setup of such a combination and whether the GHG reduction requirements should be shared equally between the two sides or if a certain split should be applied. However, a combination of efforts between supply- and demand side is likely to achieve the 1.5 °C

climate target at a quicker rate as the reduction requirements, to a larger extent, can be based on existing or nearly operational supply-side technology measures and demand-side mitigation measures. Indeed, the actual supply- and demand-side measures must be specialised based on regional steel flow and technical features. The corresponding region-specific priorities are suggested as follows:

(1) *Harnessing emerging low-carbon technologies in emerging steel markets.* As our analysis indicates, global steel producers have actively made progress on the innovation and adoption of emerging technologies[31]. Again, the effectiveness of low-carbon technologies on climate mitigation lies in their

spatial-temporal race with steel flow growth. In the short-term, there will be a faster increase of steel flows in the first 15 years (i.e., ~400 Mt additional capacity for largely primary production mainly in India, Developing Asia, and Middle East, etc.). This makes the period from 2020 to 2035 and those regions more critical. According to technical review studies[4,31,52], promising nearly zero-carbon technologies under options a) and b) (e.g., projects like HYBRIT, SALCOS, SIDERWIN, MOE in Table S4.4) will not be fully commercially viable until then. This calls for other strategies and technologies to be used in the meantime. Technologies in option c) (i.e., CCUS with direct/smelting reduction) become preferable (compared to CCUS on thermal power plants[53]) to cover steel flows in emerging steel markets before 2035. In addition, projects, such as HIsarna/FINEX/HYL with CCS, Al Reyadah CCS in Table S4.5 are nearly at commercial scale. Based on the latest IEA report[31] and others[54,55], this is particularly important for India, which is the largest emerging steel market and sitting at the next carbon emissions frontier. Moreover, there is a regional mismatch of technology innovation and implementation. At present, the EU is pioneering innovation and testing of low-carbon technologies. Hence, it is recommended to focus on and incentivise technology sharing among regions to facilitate penetration of emerging low-carbon technologies (from Tier 1 regions) in emerging steel markets (e.g., Tier 3 regions). Moreover, technology designers should consider the future resource availability (e.g., coal, natural gas, hydrogen availability) and other market factors associated with the emerging markets to aid penetration in emerging markets.

(2) *Radical early retirement of primary production capacity in China.* The existing long-lived production facilities can pose a grave threat to the 1.5 °C climate target[13]. As calculated by IEA, the committed direct $CO_2$ emissions from existing steel facilities can reach ~65 Gt from 2019 to 2060, which can nearly exhaust the remaining carbon budget in the 1.5 °C scenario. Thus, a radical early retirement of primary production capacity is needed. This will be challenging as the global blast furnace fleet is relatively young with an average age of 13 years. Our further MFA (Fig. 5a) suggests that China should reduce its primary facilities by ~170 Mt (equivalent to the present total primary production capacity in EU and North America) in the next 15 years and further ~500 Mt by 2050. China has shown the ability and willingness for early retirement of facilities, and ~100–150 Mt of existing facilities have already been retired during 2016–2020[56]. Aside from China, the excessive capacity has been widely considered as one of the main challenges facing the global steel sector at present[57] (i.e., ~30% (550 Mt) capacity redundancy in 2018), which calls for optimising the production capacity at a global scale to eliminate their low-efficiency facilities. Other mitigation options in option e) (e.g., BF top gas recycling, fuel switching, carbon circulation, etc,) should also be applied in the refurbishment and retrofitting of existing production capacities.

(3) *Towards a closed-loop steel cycle in developed nations.* Steel is, in principle, infinitely recyclable[33]. Use of scrap steel for steel production entails much lower energy consumption and GHG emissions compared to ore-based primary steel production[9,58]. This makes steel recycling an important deep-decarbonisation strategy[6,11,17,29,59]. We predict that the future global scrap supply will rise by ~3.5-fold from 2020 to 2050, and find that developed regions such as Europe, developed Asia and North America could generate scrap equivalent to their steel demand by 2050. In principle,

this can allow these regions to operate in closed steel cycles by shifting to a scrap-based EAF production route[9]. In particular, China will become the largest scrap supplier after 2035, which can enable a boost in scrap-based steel production capacity. This can substantially contribute to the decarbonisation of the global steel industry, especially if the electricity used in scrap-based EAFs is based on renewable sources, such as the case in option f (e.g., the Nucor plant in Missouri). However, in practice, the success of scrap recycling is dependent on other factors, such as social behaviour, governmental regulation, product design, and existing facility inertia. Moreover, the scrap quality from contaminated scrap mix[60] remains a great challenge for producing high-quality steel that is comparable to the primary route. Thus, closing the steel cycle requires more attention to the development of smart and low-carbon sorting, separation and refinery production, etc. as well as measures for improving source separation of steel scrap to improve the overall quality of secondary steel production.

(4) *Incorporating material efficiency measures into the decarbonisation portfolio.* Material efficiency[61] refers to increasing the output per material input. Material efficiency measures have been widely examined for steel[29,61,62] as this is considered an essential decarbonisation strategy[28]. Material efficiency measures alone are insufficient for achieving net-zero targets and must be combined with the adoption of low-carbon production technologies. Here, material efficiency can help lighten the load on the technological shift. Indeed, the latest IEA report[31] estimates that ~40% of the cumulative emissions reduction can be achieved through material efficiency (detailed strategies for each end-use sector are summarised in Table S4.5). Apart from direct material demand, those material efficiency strategies can also bring various co-benefits for systematic GHG reduction. For instance, development of lighter vehicles can reduce steel requirements by a factor of four and significantly increase fuel efficiency, thus reducing fuel use and associated GHG emissions while still maintaining the same mobility service[63].

(5) *Global cooperation for a green steel market.* As the window of opportunity for achieving the 1.5 °C target is narrowing[29], there is an urgent need for immediate and significant actions as suggested by various studies[64]. However, these cannot be functional without a joined global effort, since steel products are produced and traded in an extremely competitive global market. Indeed, the production cost of steel is expected to increase by 20–40% with those emerging carbon-free routes[20], and this may hinder the incentives for steel producers to follow decarbonisation paths, especially in developing regions with large emerging steel demand. Thus, the global innovation system for low-carbon technology development should be strengthened with regards to reducing technological costs. Moreover, the global changes in production flows can generate local and global carbon benefits (e.g., Australia as a future centre of ironmaking with significant iron ore and renewable potential[65]). Simultaneously, a global market for green steel should be fostered under various international trade and climate agreements. This could, for instance, be facilitated through certification schemes with a transparent carbon footprint from steel producers (e.g., environmental product declaration). Another suggested option for moving the steel industry toward the 1.5 °C target is high carbon taxation (i.e., 100 $/tCO_2 + 4%$ per year since 2020)[17], which would also incentivise steel production to reduce GHG emissions.

In conclusion, our study demonstrates that the total carbon reduction relies not only on the implementation of low-carbon technologies but also on their interplay with the changes of production flows at global level. Historical evidence from the steel industry shows that regional process efficiency improvement efforts have not been able to keep up with the growth in production flow, leading to a 17-fold net increase in annual GHG emissions during the studied period. Moreover, we also see that the GHG intensity of steel production at global scale has stagnated in past decades. Thus, it is recommended that key nations and steel producers clarify and develop roadmaps for steel decarbonisation which combine both supply-side and demand-side measures to stop the stagnation and further increase process efficiency as well as reduce the growth in steel demand to be able to sufficiently reduce GHG emissions at global scale. Indeed, it is important that these roadmaps are ambitious enough to meet the steel industry's targets for realising the 1.5 °C target that was set out in the Paris agreement on climate change.

## Methods

**Historical GHG emission from steel production**. The schematic diagram of Fig. S1.1 outlines the key procedures and their linkages of our analytical framework, the detailed description of which can be found in Section S1. In accordance with IPCC[66] as well as other studies[29,67], we performed a process-based approach to quantify the total greenhouse gas (GHG) emission of the iron and steel industry (assumed to be the sum of studied processes). Notably, unlike some studies that only focused on steelmaking (IPCC[66]), our analysis expanded the system boundary to the entire steel production chain as illustrated in Fig. S1.2. This allows for a more comprehensive investigation, which includes mining, material preparation, ironmaking, steelmaking and steel finishing. Those processes are represented by 19 types of dominant production technologies, the system boundary of which is presented in Fig. S2.6–S2.19. In general, our quantitative method is summarised as below:

$$TotE(t) = \sum_i E_i(t) \times TP_i(t) \qquad (1)$$

where $TotE(t)$ is the total emissions from global steel production at a studied time $t$, calculated as the sum of emissions from individual steel production technologies $i$, equal to their annual production output $TP_i(t)$ (quantified using material flow analysis) times their GHG emissions intensities $E_i(t)$ at time $t$ (quantified using LCA). This approach involves five main steps:

(1) *Production technologies investigation*. This step began with a literature review on global steel production technologies and routes change. Some of the literature, especially that on the historical development of ironmaking[27] and steelmaking[26] technologies, provides important information for our analysis, helping us define system boundaries, identify key technologies for global steel production and track technology progress in quantifying production activities and emission trends (details in Section S1).

(2) *Material flow analysis*. We carried out a dynamic material flow analysis (MFA)[68] to quantify (i) yearly production of the studied technologies and (ii) dynamic material stocks and flows along the steel cycle from the production stage to downstream stages including manufacturing, in-use and end-of-life. Details and data sources are given in Table S2.1. Such analysis can provide the activities data of each technology (process) needed for quantifying the total GHG emissions. However, all material flows are expressed as average Ferrous (Fe) content of the product amount. This can have a slight effect on the final results as presented in Section S3.1.

(3) *Emission quantification*. Following existing approach[29,67], we quantified both direct and indirect GHG emissions per mass unit output (in Fe content) on an annual basis for each steel production technology. The GHG emissions inventory we compiled draws primarily on unit process data from the Ecoinvent v.3[69,70], but also incorporates data from a variety of sources including technical reports, published LCA datasets and existing literature (for details, see Section S2.2). Notably, we assumed the technology-based GHG emissions intensity to represent the global average level while the geographical differences were ignored due to a lack of detailed and complete regional datasets. The historical development in GHG emissions intensities is based on a historical technology investigation (Section S1.2). The GHG emissions estimates have been cross-checked with the world steel association's available statistics. The historical change of each technology's emissions intensity and their variance is presented in Fig. S2.6. The GHG emissions have been grouped into three scopes: Scope 1 covers direct GHG emissions from the production site, Scope 2 covers indirect GHG emissions from the generation of electricity and heat that are used in the production, and Scope 3 covers GHG emissions associated with all other activities related to steel production (our treatment of process off-gases is similar to the previous studies[29,67]). As both electricity and heat are important inputs for steel processing, the historical development in

the distribution of energy carriers used for electricity and heat generation was also considered and assumed to follow the global average trend (see Supplementary Information 1 Tables S2.5 and S2.6). Data pertaining to the life cycle inventory data for each technology is given in Supplementary Data 1.

(4) *Decomposition analysis*. We applied the Logarithmic Mean Divisia Index (LMDI) decomposition method[71] to quantify the influence of changes in production volume and GHG intensity for the two main production routes on the absolute emission change. The total emissions, production activities and emissions intensity for each production route are obtained by aggregating the detailed data of each production technology over every five-year period from 1900 to 2015. Details on the decomposition analysis can be found in Section S4.2.

(5) *Uncertainty analysis*. Uncertainties of our analysis are mainly related to incomplete knowledge and model assumptions about the historical development and efficiencies of steel production technologies. Here, we applied the Pedigree-matrix approach[72] to derive quantitative uncertainty estimates based on qualitative data quality indicators (DQIs) which include reliability, completeness, temporal correlation, geographical correlation and technological correlation. Scores between 1 and 5 for each of the five DQIs are used to assign empirically based coefficients of variation to the parameter data. The resulting uncertainties of input data sources (see Section S2.3) were then applied in Monte-Carlo simulation (100,000 iterations) to quantify the uncertainties of model results (details in Section S3).

(6) *Model validation*. The model-simulated historical production of steel differs from historical statistics by less than 19% in most years (Fig. S3.2 and Fig. S3.3) and a Pearson correlation coefficient of 0.9997 was found between the two. We also compared our GHG intensities with previous estimates found in the literature (Section S3.3) for the period after 1950 due to a lack of earlier estimates. Ours are close to previous estimates (Fig. S3.8) with a Pearson correlation coefficient of 0.856.

**Regional retrospective and prospective analysis**. We further divided global steel stocks and flows into 8 regions (i.e., Europe, North America, Developed Asia and Oceania, China, India, Developing Asia and Middle East, Latin America and Caribbean and Africa). We applied the method developed by Pauliuk et al.[9,32] for our regional retrospective (1995–2015) and prospective analysis (2016–2050) with two updated datasets from world steel yearbooks[73] and IEA's latest projections[31] of regional steel production (i.e., Stated Policies Scenario and Sustainable Development Scenario). The Sustainable Development Scenario incorporates implementation of various material efficiency strategies which help to reduce 20% of total steel demand compared to the Stated Policies Scenario. The detailed results of each region are mapped in Figs. 2 and 5 and described in Section S4.3. We further collected datasets regarding regional crude steel production capacity from an OECD database[57]. For simplicity, the steel trade flows are not considered in our regional analysis. Thus, our analysis of steel stocks and flows should not be viewed as actual trends but as a what-if analysis on future potential trends.

**Scenario and strategies analysis**. We conducted a scenario analysis of the required efficiency improvement trends based on the projections of global steel production flows under the 1.5 °C scenario (1.5DS) carbon budget. In total, six types of scenarios were generated (details in Section S4.4). Scenario 1(S1) tests the GHG emissions growth assuming that emissions intensity continues to stagnate with IEA Stated Policy Scenario[31] demand trend, Scenario 2(S2) tests the impact of technical efficiency improvement on future GHG emissions under IEA's Sustainable Development Scenario with 30% intensity reduction by 2050[31]. Scenario 3(S3) tests the impact of material efficiency improvement on future GHG emissions under IEA's Sustainable Development Scenario with 12% reduction of total steel demand from 2019 to 2050[31]. We found that Scenario 1–3 will fully exhaust the 1.5DS carbon budget before 2050. Consequently, we further proposed Scenario 4 (S4) and Scenario 5(S5) to explore the potential of technical efficiency improvement (i.e., 0.85 tCO2-eq/t steel reduction per decade) and material efficiency improvement (i.e., 34% additional demand reduction compared to IEA Sustainable Development Scenario) under such a budget constraint, respectively. The total 1.5DS budget was estimated to be ~420–580 Gt CO2 according to reference[13], and this study adopted a high variance estimation of 106 Gt CO2 allocated by IEA 1.5DS (see Section S4.4), which accounts for 18–28% of the total budget. However, the present emissions of steel industry only account for 7–9% of global total emission, indicating a more stringent carbon constraint on future steel production than our analysis. To enrich our analysis, we further collected detailed break-through low-carbon technologies as well as detailed material efficiency strategies from various reports and studies as listed in Tables S4.4 and S4.5, respectively.

**Reporting summary**. Further information on research design is available in the Nature Research Reporting Summary linked to this article.

## Data availability

The authors declare that the source data supporting the findings of this study are study are available within the paper, and its supplementary information files. Supplementary

Information 1 contains supplementary methods and results. The supplementary results for material flow analysis and greenhouse gas emission are given in Section S3–S4 of Supplementary Information 1. Data pertaining to the life cycle inventory data for each technology is given in Supplementary Data 1 in excel format. Source data underlying all figures in the main manuscript are provided as a Source Data file. Correspondence and other requests for materials or data related to this study should be addressed to corresponding author M.R. upon reasonable request.

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

## Acknowledgements

P.W. and W-Q. C. acknowledge the support from the National Natural Science Foundation of China (No. 71904182 and 71961147003) and Key Program of Frontier Science of the Chinese Academy of Sciences (QYZDB-SSW-DQC012). P.W. acknowledges the support from UNSW Postdoc Writing Fellowship and CAST Young Talent Support Project.

## Author contributions

P.W., M.R., S.K. and M. H. designed the research. P.W., M.R and Y.Y. performed the analysis. K.F. contributed expertise on historical trend analysis. P.W., W-Q.C. and Y.Y. contributed regional analysis and future scenario settings. P.W. and M.R. led the drafting of this manuscript. All authors contributed significantly to the final writing of this article.

## Competing interests

The authors declare no competing interests.
