## [Peer Review File · Nature Communications]

REVIEWER COMMENTS

Reviewer #1 (Remarks to the Author):

Dear authors -

Your submission is both fascinating and frustrating. Fascinating in its analysis of the history of the iron and steel industry; the process and decomposition figures are excellent. Frustrating in that you summarily dismiss a fairly healthy debate going on in the community about the relative long term merits and contribution of demand side vs low GHG production for steel, cement, etc. In the space of a few sentences on page 13, with very little evidence, you dismiss the potential for low GHG steel production because it hasn't happened in the past, and that for netzero to be achieved everything must lean on the demand side. One can equally argue that while we have to some extent decoupled energy from GDP, there has been no historic decoupling of materials from GDP. I suggest reviewing "Material efficiency in clean energy transitions" (International Energy Agency (IEA), 2019). They suggest that material efficiency will only get us about 25-40% declines in steel use for a given manufacturing or infrastructure application, not counting for growth going forward. If we are to achieve netzero GHG emissions, what will take up the slack? Some very low and zero emission production methods (e.g. hydrogen DRI EAF, based on NG DRI EAF, and HISARNA plus CCS) are already beyond the pilot phase, heading towards commercial (Bataille, 2019; Fishedick et al., 2014; Gielen et al., 2020; Vogl et al., 2018), especially the Hybrit project. To make your case, you need to lean more on (Tong et al., 2019) for the argument that existing capacity will slow uptake of new technologies.

Your article opens really well. If you instead took a more neutral approach, that both demand reduction and low GHG production will be needed, the paper would read more reasonably.

Sincerely,
Your reviewer

References

Bataille, C., 2019. Physical and policy pathways to net-zero emissions industry. *WIREs Wiley Interdisciplinary Reviews* 1–20. doi:10.1002/wcc.633

Fishedick, M., Marzinkowski, J., Winzer, P., Weigel, M., 2014. Techno-economic evaluation of innovative steel production technologies. *Journal of Cleaner Production* 84, 563–580. doi:10.1016/j.jclepro.2014.05.063

Gielen, D., Saygin, D., Taibi, E., Birat, J.P., 2020. Renewables-based decarbonization and relocation of iron and steel making: A case study. *Journal of Industrial Ecology* 1–13. doi:10.1111/jiec.12997

International Energy Agency (IEA), 2019. Material efficiency in clean energy transitions [WWW Document]. doi:10.1787/aeaaccd8-en

Tong, D., Zhang, Q., Zheng, Y., Caldeira, K., Shearer, C., Hong, C., Qin, Y., Davis, S.J., 2019. Committed emissions from existing energy infrastructure jeopardize 1.5 °C climate target. *Nature* 3. doi:10.1038/s41586-019-1364-3

Vogl, V., Åhman, M., Nilsson, L.J., 2018. Assessment of hydrogen direct reduction for fossil-free steelmaking. *Journal of Cleaner Production* 203, 736–745. doi:10.1016/j.jclepro.2018.08.279

Reviewer #2 (Remarks to the Author):

Comments

I must say that this is a good research with intensive man-hour efforts and the research is carefully designed and presented clearly in general. The transparency of the data source and method is also solid. I appreciate the authors for their work and believe it will add value to the academic field. The authors examined the evolution of steel industry over the past century, and it is useful, because 'studying the history is the passport to the future'. Generally, I recommend an acceptance for publication after addressing the major concerns as follows.

Introduction

It is better to highlight the new knowledge in this study as compared with the previous, such as the ref. 1 and 2, that 'To achieve a climate-safe future, the iron and steel (steel, hereafter) industry must reduce 50-85% of its emissions by 2050'.

Methodology section:

- (1) I think a schematic diagram of method will be helpful to understand the methodology clearly and systematically. Presently, the method section is broken down into different parts and it is hard to have a whole picture of the method.
- (2) There are intensive assumptions in the method section, and it will be better to add some words to solidify the reasonability of such assumptions.
- (3) Clarify the system boundary of LCA for each technology.
- (4) Most of the scenarios followed previous studies. I was wondering the novelty and major differences for your three scenarios as compared with the six scenarios in other studies.

Policy implications.

- (1) Demand-side mitigation strategies are the key policy implications proposed in this study. However, the discussion on this could be enriched. Presently, the discussion on demand-side mitigation strategies reads some superficial. Four sectors, construction, transportation, machinery, and consumer goods, consumed large part of steel product. Thus, more attention should be paid, from the demand-side, to finger out the appropriate mitigation measures.
- (2) The key finding of the research is that "we found that the GHG emission intensity of the global steel industry has stagnated in the recent 15-20 years. This stagnation indicates that further improvement of process efficiency is difficult and costly, suggesting that demand-side solutions are urgently needed to achieve climate targets". --- This is important from the perspective of temporal resolution to uncover the stagnation of process efficiency globally. However, this is the general view of the steel industry while questing for decarbonation. My suggestion is that it will be much better to propose a temporal-spatial resolution stagnation of process efficiency, thus, the policy implication will hit the point much clear. Presently, based on temporal resolution, the demand-side mitigation strategies as recommended in the manuscript is some general. We still do not know where to start from the production side and demand side.

Some detailed comments

- (1) L71-72, of which around half pertained to the open-hearth furnace (Fig.1b) ---- Fig. 1b should be Fig. 1a.
- (2) L74, GHG emissions from mineral treatment were ~22 Gt CO₂-eq --- my question is that the mineral treatment process and the number (22 Gt) are not clearly shown in the Fig. 1 or Fig. 2.
- (3) L75, add (see Fig. S1.1) behind 'two major production routes'.
- (4) L78-79, 'Historically, it is estimated the primary production route has emitted ~132 Gt CO₂-eq, accounting for over 90% of total GHG emissions'--- I think it will be better to add the emission ratio of primary route and second route at present status, such as in the year of 2015. Because the second route is more efficient and less carbon intensive as compared with the primary route. I noticed in L95-96 that the share of production with the secondary route is about 20%.
- (5) L89-90, I think the (Fig. 3h) and (Fig. 3i) are misplaced, perhaps it should be (Fig. 3e) and (Fig. 3h), respectively. Also, in L96, I am afraid the (Fig. 3d) is incorrect. Please check the citation of Fig. 3 (a-i) in the manuscript.
- (6) L125-126, 'However, there was no fundamental technology replacement occurring in past 2-3 decades.'--- I think it will be better to add more words on the reasons.
- (7) L140, (Fig. 4a-c) --- I think it should be (Fig. 4a, c, e). Moreover, it seems the Fig. 4a and Fig. 4c are the same. Please check it.
- (8) L146, add (Fig. 4c, e) after 'remained at ~3.0 t CO₂-eq/t steel and ~1.0 t CO₂-eq/t steel'
- (9) The use of Fig. S2.2B is not explained in the maintext.

(10) L148-150, the authors claimed that 'Steel industries in different countries have introduced policies, measures, and innovations to improve process efficiency (S4.4) but they have been insufficient to reduce the GHG intensity of steel production on a global level.' --- I think such claim is very general. It will be much better to analysis the new production lines employed after 1995 or even recent to find out their energy efficiencies and GHG emission intensities. The manuscript made a cumulative analysis of the GHG emissions; thus, it will perhaps mislead the reader that the newly employed production lines are also carbon intensive. Because as I know, a large number of short processes steelmaking (the secondary route) are established in the new century.

(11) L157, the citation of (Fig. 3f) is incorrect. It may be (Fig. 4f).

(12) L151-167, the authors focused on efficiency improvement in steel production. It is good. I think there is one more aspect should be added, the structure optimization. It means the scale-technology structure of newly employed facilities, along with the retirement of small scale and low efficiency facilities. with the increase of large scale and high efficiency facilities start working, the overall intensity of GHG emission and energy efficiency of steel making industry are no doubt changed. The structure optimization is also helpful for the Future 1.5 °C climate targets as discussed in L177-189.

(13) Another issue related to the structure optimization is the redundancy of steelmaking production capacity, such as in China. This comment is an echo of my previous one on the temporal-spatial analysis of the steelmaking industry globally. I suggest the authors add more words on the analysis of excessive capacity and the loading of in-use stocks of steelmaking facilities by targeting the mitigation strategies.

(14) The readability and resolution of FIg. 2.3 A could be improved, and my concern is that the steel product mass allocation method is not self-explained. Additional explanation is necessary to make it clear. How do you come to the percentiles in the section A?

(15) L223, the citation of (Fig. 4c) is incorrect. It is the same for (Fig. 4b) in L226.

Detailed comments on the supplementary materials.

(1) SI, L224-226, two symbols, t and t , are not defined. ν is not included in the equation.

(2) SI, L306-308, the presentation of parameters is not consistent with the equation S5.

(3) Figure S3.8, the color difference between the legends of upper bond value and lower bond value are very minor, it is better to make them different.

Response to reviewers' comments on the manuscript:

Efficiency Stagnation in Global Steel Production Urges Joint Supply- and Demand-side Mitigation Efforts

Dear reviewers,

We would like to thank you for your time and effort in reviewing this manuscript and for providing very valuable and relevant comments on the manuscript to help augmenting its quality. Your comments are addressed point-by-point in the following (in blue text and with quotations of changed text wherever relevant).

Sincerely,

Morten Ryberg (on behalf of all co-authors)

Reviewers' comments:

Response to Reviewer #1

Comment#1.1

Dear authors

Your submission is both fascinating and frustrating. Fascinating in its analysis of the history of the iron and steel industry; the process and decomposition figures are excellent. Frustrating in that you summarily dismiss a fairly healthy debate going on in the community about the relative long-term merits and contribution of demand-side vs low GHG production for steel, cement, etc.

Response to reviewer:

We are highly appreciative of your encouraging comments and insightful suggestions. Your suggestion on the healthy debate benefited us a lot. Accordingly, we have expanded our study to take into account measures related to the supply side, such as low GHG production of steel. In doing so, we have added an updated literature review on low GHG production technology options in Table S4.4 and on material efficiency strategies in Table S4.5. We use the review of the recent development on the supply- and demand side to deepen the analysis of the “interplay between material flows and supply-side technical efficiency”. Here, we argue an urgent need to combine strategies from both sides to achieve deep carbonization of steel and other “difficult-to-mitigate” sectors. Thus, we have also substantially revised this part of the discussion in the manuscript.

Specific changes to manuscript: The manuscript has been revised throughout to reflect both the supply and demand side of steel production and the different options for reducing GHG emissions from both sides.

Comment#1.2

In the space of a few sentences on page 13, with very little evidence, you dismiss the potential for low GHG steel production because it hasn't happened in the past, and that for net-zero to be achieved everything must lean on the demand side.

Response:

As indicated before, we agree with your comments. We have revised this part to provide a deeper analysis of the potentials for low GHG steel production. Through our regional analysis, we have also shown that there have been important technological improvements since 1995. However, the benefits of these have largely been offset by the historical growth of steel production with low process efficiency (mainly in China, India, and other developing nations). Thus, if there had not been an improvement in steel production efficiency, total emissions would have been much higher. Accordingly, this further underlines the need for the implementation of technological improvement as well as demand-side strategies for future GHG reduction in the steel industry.

Specific changes to manuscript:

Page 11 Line 189:

During 1995-2015, steel industries in different regions have introduced various commitments, measures, and innovations to improve process efficiency. For instance, Europe, as one of the most efficient steel producing regions, has made various efforts (e.g., ULCOS program started in 2004⁴²) to reduce its carbon intensity by nearly 50%^{43,44}. Efforts have also been made in Japan (e.g., the COURSE50 project started in 2008⁴⁵), USA (30% energy intensity reduction since 1990⁴⁶), and other nations. With the help of continuous technological advancements and early retirement of redundant inefficient facilities, China has also made progress in improving its steel production efficiency (by ~30%⁴⁷) during the past few decades. Moreover, the state-of-the-art ironmaking technology blast furnace is approaching the practical minimum energy requirement³¹. Nevertheless, those regional technical improvements are found to be insufficient to reduce the overall GHG intensity (let alone the total GHG emission) of steel production at global level (as indicated in Fig. 3b-c) due to the structure changes in regional production flows.

By categorizing those regions into different groups based on their emissions intensities (i.e., Tier 1, 2, and 3 in Fig. 3b), we found an 8-fold expansion of crude steel production flows from the most carbon-intensive regions (i.e., Tier 3; Fig.3c), rising from 129 Mt/year in 1995 to 914 Mt/year in 2015. By contrast, the operating production capacity from low (Tier 1) and medium (Tier 2) carbon-intensive regions has shrunk in the same period (Fig.2b). Indeed, the share of global steel production in Tier 1 and 2 regions decreased from 83% to 43% during the studied period. The change in production flow structure has offset the mentioned regional technical advances. Notably, the poor carbon performance of Tier 3 regions was not simply attributed to their technical backwardness³⁸, but also closely linked to the distribution between primary and secondary production routes. Emerging economies have been fuelled by the fast expansion of steel flows (production and consumption) and limited scrap availability. Thus, the production was dominated by the primary production route, which resulted in a high GHG intensity for the entire steel industry (see the comparison between USA and China^{48,49}). Accordingly, our results highlight the necessity of coordinating the implementation of low-carbon technologies while also considering the regional

structural changes in steel flows (particularly for emerging economies) to achieve deep decarbonization of global steel production.

Comment#1.3

One can equally argue that while we have to some extent decoupled energy from GDP, there has been no historic decoupling of materials from GDP. I suggest reviewing “Material efficiency in clean energy transitions” (International Energy Agency (IEA), 2019). They suggest that material efficiency will only get us about 25-40% declines in steel use for a given manufacturing or infrastructure application, not counting for growth going forward.

Response:

We fully agree. The “Material efficiency in clean energy transitions” IEA report, together with the latest IEA report on the steel industry (<https://www.iea.org/reports/iron-and-steel-technology-roadmap>), has been carefully reviewed. Thus, we agree that our original statement about the importance of material efficiency was overstated. We have moderated it in the revised version and adopted a more neutral stance on the importance of supply- and demand-side strategies for achieving GHG reductions.

Specific changes to manuscript:

Page 13 Line 229:

It will require a rapid innovation and implementation of low-carbon technologies to achieve the necessary reduction in GHG intensity. We summarized 37 types of breakthrough technologies in Section S4.5 and grouped them into seven categories: (a) Hydrogen-based options, (b) Electrolysis-based options, (c) CCUS with direct/smelting reduction, (d) Biomass-based options, (e) Blast furnace-improvement, (f) Carbon-free EAF, and (g) Low-carbon rolling technologies. In combination, the breakthrough technologies have the potential to reduce GHG emissions at the required rate to become carbon-neutral by 2047. However, the rate of technology development and implementation is critical. To realize the 1.5 °C climate target, breakthrough technologies must be developed to a level that is fully operational and be implemented at global scale. Based on the review of the 37 breakthrough technologies, the rate of development appears too slow as most technologies are planned to be available in 10-25 years and with only limited implementation at that time. Aside from supply-side technology measures, various demand-side mitigation measures targeting material flows and technically assisted lifestyle change beyond the direct control of the industry have received attention ^{25,52,53}.

Given the pressing carbon constraints on steel industry, we argue the need for integration of supply-side and demand-side measures to meet the 1.5 °C climate target. Indeed, a combination of supply- and demand side measures would entail less radical reduction measures on both sides compared to if one of the sides would need to achieve the reductions alone. We will not venture into the setup of such combination and whether the GHG reduction requirements should be shared equally between the two sides or if a certain split should be applied. However, a combination of efforts between supply- and demand side is likely to achieve the 1.5 °C climate target at a quicker rate as the reduction requirements, to a larger extent, can be based on existing or nearly operational supply-side technology measures and demand-side mitigation measures. Indeed, the actual supply- and demand side measures must be specialized based on regional steel flow and technical features.

Page 16 Line 316:

Material efficiency⁶⁴ refers to increasing the output per material input. Material efficiency measures have been widely examined for steel^{29,64,65} as this is considered an essential decarbonization strategy²⁸. Material efficiency measures alone are insufficient for achieving net-zero targets and must be combined with the adoption of low-carbon production technologies. Here, material efficiency can help lighten the load on the technological shift. Indeed, the latest IEA report³¹ estimates that around 40% of the cumulative emissions reduction can be achieved through material efficiency (detailed strategies for each end-use sector are summarized in Table S4.5).

Comment#1.4

If we are to achieve net-zero GHG emissions, what will take up the slack? Some very low and zero emission production methods (e.g. hydrogen DRI EAF, based on NG DRI EAF, and HISARNA plus CCS) are already beyond the pilot phase, heading towards commercial (Bataille, 2019; Fishedick et al., 2014; Gielen et al., 2020; Vogl et al., 2018), especially the Hybrit project. To make your case, you need to lean more on (Tong et al., 2019) for the argument that existing capacity will slow uptake of new technologies.

Response:

Thanks for your insightful suggestions and references. We have added a discussion of five options that combine supply and demand side mitigation strategies to move towards net-zero GHG emissions. In particular, the work of Tong et al. (2019) has been incorporated to improve the discussion of existing capacity and its redundancy and the work by Gielsen et al. (2020) was used to elaborate the discussion of climate-driven global steel flow optimization.

Specific changes to manuscript: Please see the new discussion part starting from Page 15 Line 267. Here we discuss 5 options that combine supply and demand-side mitigation strategies.

The 5 options are:

- 1) Harnessing emerging low-carbon technologies in emerging steel markets
- 2) Radical early-retirement of primary production capacity in China
- 3) Towards a closed-loop steel cycle in developed nations
- 4) Incorporating material efficiency measures into the decarbonization portfolio
- 5) Global cooperation for a green steel market

Comment#1.5

Your article opens really well. If you instead took a more neutral approach, that both demand reduction and low GHG production will be needed, the paper would read more reasonably.

Response:

Thank you for your comments about this. We fully agree and we hope our revisions have ensured a more neutral approach on the need for both demand and supply-side reduction measures.

Response to Reviewer #2

Comment#2.1

I must say that this is a good research with intensive man-hour efforts and the research is carefully designed and presented clearly in general. The transparency of the data source and method is also solid. I appreciate the authors for their work and believe it will add value to the academic field. The authors examined the evolution of steel industry over the past century, and it is useful, because ‘studying the history is the passport to the future’. Generally, I recommend an acceptance for publication after addressing the major concerns as follows.

Response:

Thanks for your encouraging comments and comprehensive assessment. we particularly like the sentence “studying the history is the passport to the future”! We have revised the manuscript thoroughly to address all of your constructive comments.

Comment#2.2

Introduction

It is better to highlight the new knowledge in this study as compared with the previous, such as the ref. 1 and 2, that “To achieve a climate-safe future, the iron and steel (steel, hereafter) industry must reduce 50-85% of its emissions by 2050’.

Response:

We have updated the latest carbon emission requirement for global steel production based on the new knowledge from the recent IPCC reports and other publications.

Specific changes to manuscript:

Page 3 Line 39:

To achieve a climate-safe future as required by the Paris Agreement, there is a need for reaching net-zero emissions by around 2050 and net negative emissions thereafter ^{1,2}, for every sector including the steel industry ^{3,4}.

Comment#2.3

Methodology section:

(1) I think a schematic diagram of method will be helpful to understand the methodology clearly and systematically. Presently, the method section is broken down into different parts and it is hard to have a whole picture of the method.

Response:

We agree on this very useful comment, and have added a schematic diagram to link all parts of our method in Fig. S1.1 with detailed description in Section S1.1 of Supporting Information, and further cited this figure in the method part of the main text.

Specific changes to manuscript:

Supplementary Information 1, Page S3 Line 1:

Our analysis is performed based on the procedures given in Fig.S1.1, which include three major steps. In Section S1, we first clarify our system boundary, and explore the historical development of the studied processes (and technologies) through the compressive investigation of various technical reports and publications. Secondly, in the entire Section S2, we present detailed procedures for our material flow analysis and environmental impact analysis. This allows for generating results related to material stocks and flows throughout each studied process along the global steel cycle from 1900 to 2015, as well as the trend in greenhouse gas (GHG) emission of each studied process during 1900-2015. By combining these two sets of results, we obtain the historical trend of total GHG emissions from global steel production in the past 115 years, and perform uncertainty analysis of the results (the method is described in Section S2.3 and the results are presented in Section S3). Furthermore, in Step 3 (to enrich the implications of our results), we further deepen our analysis of the global level by looking at regional aspects. This is done by separating the World into 8 regions (i.e. Europe, North America, Developed Asia and Oceania, China, India, Developing Asia and Middle East, Latin America and Caribbean, Africa). Here, we further quantify the regional material stocks and flows since 1995, and examine their GHG emission intensity performance. Meanwhile, we also investigate different types of low-carbon technologies related to steel production and perform a scenario analysis until 2050 to provide recommendations on the required mitigation strategies for achieving climate change targets.

Fig.S1.1 Schematic diagram of our analytical methods

Comment#2.4

(2) There are intensive assumptions in the method section, and it will be better to add some words to solidify the reasonability of such assumptions.

Response:

Thank you for this comment. We have elaborated our description in the method section by further clarifying the assumptions that existed in our material flow analysis and emission quantification method.

Specific changes to manuscript:

Page 18 Line 365:

2) Material flow analysis. We carried out a dynamic material flow analysis (MFA) ⁷¹ to quantify i) yearly production of the studied technologies and ii) dynamic material stocks and flows along the steel cycle from the production stage to downstream stages including manufacturing, in-use, and end-of-life. Details and data sources are given in Table S2.1. Such analysis can provide the activities data of each technology (process) needed for quantifying the total GHG emissions. However, all material flows are expressed as average Ferrous (Fe) content of the product amount. This can have a slight effect on the final results as presented in Section S3.1.

(3) Emission quantification. Following existing approach ^{29,70}, we quantified both direct and indirect GHG emissions per mass unit output (in Fe content) on an annual basis for each steel production technology. The GHG emissions inventory we compiled draws primarily on unit process data from the Ecoinvent v.3 ^{72,73}, but also incorporates data from a variety of sources including technical reports, published LCA datasets, and existing literature (for details, see Section S2.2). Notably, we assumed the technology-based GHG emissions intensity to represent the global average level while the geographical differences were ignored due to a lack of detailed and complete regional datasets. The historical development in GHG emissions intensities is based on a historical technology investigation (Section S1.2). The GHG emissions estimates have been cross-checked with the world steel association's available statistics. The historical change of each technology's emissions intensity and their variance is presented in Fig. S2.6. The GHG emissions have been grouped into three scopes: Scope 1 covers direct GHG emissions from the production site, Scope 2 covers indirect GHG emissions from the generation of electricity and heat that are used in the production, and Scope 3 covers GHG emissions associated with all other activities related to steel production (our treatment of process off-gases is similar to the previous studies ^{29,70}). As both electricity and heat are important inputs for steel processing, the historical development in the distribution of energy carriers used for electricity and heat generation was also considered and assumed to follow the global average trend (See Table S2.5 and Table S2.6).

Comment#2.5

(3) Clarify the system boundary of LCA for each technology.

Response:

We agree that the system boundary of each technology should be clarified. We have provided flow charts indicating the system boundary for each technology in Supporting Information 1 Fig.S2.6-S2.15. Moreover, Supplementary Information 2 provides a full overview of the modeling of each technology including process inputs and outputs.

Specific changes to manuscript:

Please see Supplementary Information 1 Section S2.2.2 “System boundary of LCA for each technology” where all system boundary figures are provided.

Comment#2.6

(4) Most of the scenarios followed previous studies. I was wondering the novelty and major differences for your three scenarios as compared with the six scenarios in other studies.

Response:

We have re-conducted our scenario analysis to indicate the future pathways of required technical and material efficiency improvement to follow the 1.5DS pathway. Accordingly, we propose six new scenarios (details can be found in Supplementary Information 1 Section S4.4). Scenario 1 (S1) tests the GHG emissions growth if the emissions intensity stays stagnated with IEA Stated Policy Scenario demand trend, Scenario 2 (S2) tests the impact of technical efficiency improvement on future GHG emissions under IEA’s Sustainable Development Scenario with 30% intensity reduction by 2050, and Scenario 3 (S3) tests the impact of material efficiency improvement on future GHG emissions under IEA’s Sustainable Development Scenario with 12% reduction of total steel demand from 2019 to 2050. We have summarized the settings and findings of our scenario analysis in Table S4.2.

Table S4.2 Settings and Results of proposed future scenarios

Scenario	Settings	Implications
1. Efficiency stagnation scenario	 Production trend from IEA Stated Policy Scenario (71 Gt during 2019-50) Efficiency trend keeps stagnating 	The remaining carbon budget will be extracted by year ~2033
2. Technical efficiency improvement	 Production trend from IEA Stated Policy Scenario (71 Gt during 2019-50) Emission efficiency follows the annual rate of IEA Sustainable Development Scenario (0.71 CO₂-eq/t by 2050) 	The remaining carbon budget will be extracted by year ~2037
3. Material efficiency improvement	 Production trend from IEA Sustainable Development Scenario (64 Gt during 2019-50) Emission efficiency follows the annual rate of IEA Sustainable Development Scenario (0.71 CO₂-eq/t by 2050) 	The remaining carbon budget will be extracted by year ~2038
4. Budget-constrained technical efficiency scenario	 Production trend from IEA Stated Policy Scenario (71 Gt during 2019-50) Not extract the 1.5DS budget by 2050 	Radical GHG intensity decrease to zero by 2046 at an average rate of 0.85 t CO ₂ -eq/t steel per decade

5. Budget-constrained material efficiency scenario	 Emission efficiency follows the annual rate of IEA Sustainable Development Scenario (0.71 CO₂-eq/t by 2050) Not extract the 1.5DS budget by 2050 	If GHG intensity decrease to 0.71 t CO ₂ -eq/t by 2050 (similar to scenario 3), and the total demand should be cut by additional 34% (43 Gt during 2019-50) compared to IEA Sustainable Development Scenario
--	--	---

We find that Scenario 1-3 will fully exhaust the 1.5DS carbon budget before 2050. Consequently, we further propose Scenario 4(S4) and Scenario 5(S5) to explore potential needs for technical efficiency improvements (i.e. 0.85 tCO₂-eq/t steel reduction per decade) and material efficiency improvements (i.e. 34% additional demand reduction compared to IEA Sustainable Development Scenario) under such budget constraints, respectively. Consequently, we have redrawn Fig.5 to clarify our results.

Comment#2.7

Policy implications.

(1) Demand-side mitigation strategies are the key policy implications proposed in this study. However, the discussion on this could be enriched. Presently, the discussion on demand-side mitigation strategies reads some superficial. Four sectors, construction, transportation, machinery, and consumer goods, consumed large part of steel product. Thus, more attention should be paid, from the demand-side, to finger out the appropriate mitigation steel measures.

Response:

We agree on this useful comment and have reviewed six groups of strategies, i.e., Less Material Same Service, More Intensive Use, Lifespan Extension, Fabrication Scrap Diversion, Reuse of End-of-Life Scrap, and Yield Improvement) for the four end-use sectors based on the IEA steel report ¹ and other publications in Table S4.5. Accordingly, in the discussion and scenario analysis part of the main text, we have explored the impact of material efficiency based on the detailed implementation route already proposed by the latest IEA steel report and further explored their

¹ IEA. Iron and Steel Technology: Towards more sustainable steelmaking. Paris, France: 2020. <https://doi.org/10.1016/B978-0-08-096988-6.00001-8>.

impacts on the achievement of the 1.5 °C target. Based on this, we have enriched our policy implications and recommendations on measures for reducing GHG emissions. Here, we have added a discussion of five options that combine supply and demand side mitigation strategies to move towards net-zero GHG emissions.

Specific changes to manuscript: Please see the new discussion part starting from Page 15 Line 267. Here we discuss 5 options that combine supply and demand side mitigation strategies.

The 5 options are:

- 1) Harnessing emerging low-carbon technologies in emerging steel markets
- 2) Radical early-retirement of primary production capacity in China
- 3) Towards a closed-loop steel cycle in developed nations
- 4) Incorporating material efficiency measures into the decarbonization portfolio
- 5) Global cooperation for a green steel market

Comment#2.8

(2) The key finding of the research is that “we found that the GHG emission intensity of the global steel industry has stagnated in the recent 15-20 years. This stagnation indicates that further improvement of process efficiency is difficult and costly, suggesting that demand-side solutions are urgently needed to achieve climate targets”. --- This is important from the perspective of temporal resolution to uncover the stagnation of process efficiency globally. However, this is the general view of the steel industry while questing for decarbonation. My suggestion is that it will be much better to propose a temporal-spatial resolution stagnation of process efficiency, thus, the policy implication will hit the point much clear. Presently, based on temporal resolution, the demand-side mitigation strategies as recommended in the manuscript is some general. We still do not know where to start from the production side and demand side.

Response:

We fully agree with you about assessing the temporal-spatial development and really appreciate this useful and insightful suggestion. Therefore, we have made an effort to explore the temporal-spatial resolution stagnation of process efficiency by separating the world into 8 regions (i.e. Europe, North America, Developed Asia and Oceania, China, India, Developing Asia and Middle East, Latin America and Caribbean, Africa). Moreover, we have improved our method to explore the steel stocks and flows from 1995 to 2050. Based on our regional analysis, it is interesting to find that there were technology improvements across the globe during the past few decades. However, these improvements were offset by the concomitant growth in steel production with low process efficiency (mainly in China, India, and other developing nations).

Again, we really appreciate your suggestion as the additional work we have conducted from the temporal-spatial perspective has greatly strengthened our analysis and enriched our findings,

e.g., for gauging the true improvement potentials and enabling more targeted regional recommendations reduction measures.

Specific changes to manuscript:

Page 11 Line 189:

During 1995-2015, steel industries in different regions have introduced various commitments, measures and innovations to improve process efficiency. For instance, Europe, as one of most efficient steel producing regions, has made various efforts (e.g., ULCOS program started in 2004 ⁴²) to reduce its carbon intensity by nearly 50% ^{43,44}. Efforts have also been made in Japan (e.g., the COURSE50 project started in 2008 ⁴⁵), USA (30% energy intensity reduction since 1990 ⁴⁶), and other nations. With the help of continuous technological advancements and early retirement of redundant inefficient facilities, China has also made progress in improving its steel production efficiency (by ~30% ⁴⁷) during the past few decades. Moreover, the state-of-the-art ironmaking technology blast furnace is approaching the practical minimum energy requirement ³¹. Nevertheless, those regional technical improvements are found to be insufficient to reduce the overall GHG intensity (let alone the total GHG emission) of steel production at global level (as indicated in Fig. 3b-c) due to the structure changes in regional production flows.

By categorizing those regions into different groups based on their emissions intensities (i.e., Tier 1, 2, and 3 in Fig. 3b), we found an 8-fold expansion of crude steel production flows from the most carbon-intensive regions (i.e., Tier 3; Fig.3c), rising from 129 Mt/year in 1995 to 914 Mt/year in 2015. By contrast, the operating production capacity from low (Tier 1) and medium (Tier 2) carbon-intensive regions has shrunk in the same period (Fig.2b). Indeed, the share of global steel production in Tier 1 and 2 regions decreased from 83% to 43% during the studied period. The change in production flow structure has offset the mentioned regional technical advances. Notably, the poor carbon performance of Tier 3 regions was not simply attributed to their technical backwardness ³⁸, but also closely linked to the distribution between primary and secondary production routes. Emerging economies have been fuelled by the fast expansion of steel flows (production and consumption) and limited scrap availability. Thus, the production was dominated by the primary production route, which resulted in a high GHG intensity for the entire steel industry (see the comparison between USA and China ^{48,49}). Accordingly, our results highlight the necessity of coordinating the implementation of low-carbon technologies while also considering the regional structural changes in steel flows (particularly for emerging economies) to achieve deep decarbonization of global steel production.

Fig. 3 Historical steel flows and stocks along with its life cycle, and the recent trend of regional flows since 1995. a The global historical steel cycle in a Sankey diagram where the numbers represent the accumulated annual flows over the past 115 years. b The change in operating production capacity for each studied region between 1995 and 2015, which is assumed equal to annual crude steel production based on data from World Steel Association yearbook 37. The CO₂ intensities of these regions are obtained based on the benchmark investigation of Global Efficiency Intelligence 38 where only a limited number of nations is collected for the comparison of the situation 39 in the year 2016. The boundary for this calculation differs from ours, so we only categorised these nations into three efficiency tiers for indicative analysis. c The annual trend of steel production in the studied regions, and demonstrates that the historical growth in production flow was driven by the emerging nations with poor efficiency performance as shown in their share change between 1995 and 2015.

Page 13 Line 241:

Given the pressing carbon constraints on steel industry, we argue the need for integration of supply-side and demand-side measures to meet the 1.5 °C climate target. Indeed, a combination of supply- and demand-side measures would entail less radical reduction measures on both sides compared to if one of the sides would need to achieve the reductions alone. We will not venture into the setup of such combination and whether the GHG reduction requirements should be shared equally between the two sides or if a certain split should be applied. However, a combination of efforts between supply- and demand-side is likely to achieve the 1.5 °C climate target at a quicker rate as the reduction requirements, to a larger extent, can be based on existing or nearly operational supply-side technology measures and demand-side mitigation measures. Indeed, the actual supply- and demand-side measures must be specialized based on regional steel flow and technical features. The corresponding regional-specific priorities are suggested as follows:

Comment#2.9

Some detailed comments

(1) L71-72, of which around half pertained to the open-hearth furnace (Fig.1b) ---- Fig. 1b should be Fig. 1a.

Response:

This has been corrected.

Specific changes to manuscript:

Page 5 Line 77:

of which around half pertained to the open-hearth furnace (Fig.1a).

Comment#2.10

(2) L74, GHG emissions from mineral treatment were ~22 Gt CO₂-eq --- my question is that the mineral treatment process and the number (22 Gt) are not clearly shown in the Fig. 1 or Fig. 2.

Response:

Thanks for your kind comment. We have made an error here. As marked in Fig. 1, the mineral treatment includes the technology of mining (5.5 Gt), sintering (12.7 Gt), and pelleting (0.5 Gt), which totaled 18.7 Gt, rather than 22 Gt (due to a mistaken inclusion of the EAF process).

Accordingly, we have corrected this error, and further improved Fig.1 to mark those numbers clearly in Fig.1.

Specific changes to manuscript:

Page 5 Line 80:

GHG emissions from mineral treatment were ~18.7 Gt CO₂-eq

Comment#2.11

(3) L75, add (see Fig. S1.1) behind 'two major production routes'

Response:

This has been added.

Specific changes to manuscript:

Page 5 Line 81:

The entire production system can be divided into two major production routes (see Fig. S1.2)

Comment#2.12

(4) L78-79, 'Historically, it is estimated the primary production route has emitted ~132 Gt CO₂-eq, accounting for over 90% of total GHG emissions'--- I think it will be better to add the emission ratio of primary route and second route at present status, such as in the year of 2015. Because the second route is more efficient and less carbon intensive as compared with the primary route. I noticed in L95-96 that the share of production with the secondary route is about 20%.

Response:

This has been added. The present share of secondary in annual emission is about 5% in 2015 as shown in Fig.S4.2.

Specific changes to manuscript:

Page 5 Line 84:

The secondary production route was around one-eighth as carbon-intensive as the primary route ³¹, and accounted for about 5% of total annual GHG emissions in 2015 as visible from Fig.2d and 2e. Historically, it is estimated that the primary production route has emitted ~132 Gt CO₂-eq over the studied period, accounting for over 90% of the total historical GHG emissions from steel production.

Comment#2.13

(5) L89-90, I think the (Fig. 3h) and (Fig. 3i) are misplaced, perhaps it should be (Fig. 3e) and (Fig. 3h), respectively. Also, in L96, I am afraid the (Fig. 3d) is incorrect. Please check the citation of Fig. 3 (a-i) in the manuscript.

Response:

Thank you for spotting these errors. We have corrected these mentioned citations of figures, and double-checked the whole manuscript to ensure all citations are correct.

Comment#2.14

(6) L125-126, 'However, there was no fundamental technology replacement occurring in past 2-3 decades.'--- I think it will be better to add more words on the reasons.

Response:

This sentence has been deleted as we think this statement was confusing and misleading, and we since further explored the reasons for this in our regional analysis section.

Comment#2.15

(7) L140, (Fig. 4a-c) --- I think it should be (Fig. 4a, c, e). Moreover, it seems the Fig. 4a and Fig. 4c are the same. Please check it.

Response:

We have corrected these mentioned citations of figures. Fig. 4a is the emissions intensity trend of total steel production, which is dominated by the primary production in Fig. 4c. That is the reason why they look similar, but not the same. This is indicated in the figure below.

Specific changes to manuscript:

Page 10 Line 155:

Notably, the GHG intensity of steel production has been stagnating at ~2.5 t CO₂-eq/t steel after 1995 (Fig. 4a, 4c, and 4e).

Comment#2.16

(8) L146, add (Fig. 4c, e) after ‘remained at ~3.0 t CO₂-eq/t steel and ~1.0 t CO₂-eq/t steel’

Response:

These have been added.

Specific changes to manuscript:

Page 10 Line 160:

remained at ~3.0 t CO₂-eq/t steel and ~1.0 t CO₂-eq/t steel (Fig. 4c, 4e), respectively

Comment#2.17

(9) The use of Fig. S2.2B is not explained in the main text.

Response:

Thank you for spotting this. We have deleted this information as it was not needed in the main manuscript.

Comment#2.18

(10) L148-150, the authors claimed that ‘Steel industries in different countries have introduced policies, measures, and innovations to improve process efficiency (S4.4) but they have been insufficient to reduce the GHG intensity of steel production on a global level.’ --- I think such claim is very general. It will be much better to analysis the new production lines employed after 1995

or even recent to find out their energy efficiencies and GHG emission intensities. The manuscript made a cumulative analysis of the GHG emissions; thus, it will perhaps mislead the reader that the newly employed production lines are also carbon-intensive. Because as I know, a large number of short processes steelmaking (the secondary route) are established in the new century.

Response:

We agree with this useful comment. Please see our response to Comment#2.8 for the explanation of how we have revised the manuscript to address this comment and the changes made to the manuscript.

Comment#2.19

(11) L157, the citation of (Fig. 3f) is incorrect. It may be (Fig. 4f).

Response:

We have corrected these mentioned citations of figures, and double-checked the whole manuscript to ensure all citations are correct.

Comment#2.20

(12) L151-167, the authors focused on efficiency improvement in steel production. It is good. I think there is one more aspect should be added, the structure optimization. It means the scale-technology structure of newly employed facilities, along with the retirement of small scale and low efficiency facilities. with the increase of large scale and high efficiency facilities start working, the overall intensity of GHG emission and energy efficiency of steel making industry are no doubt changed. The structure optimization is also helpful for the future 1.5 °C climate targets as discussed in L177-189.

Response:

Thanks for your insightful suggestions. The further facility-level analysis of each nation requires high-resolution facility data to check their scale and efficiency; such data is unfortunately not available. Thus, we have performed the historical structure change of regional flow as well as its emissions intensity (see section “Regional technical efforts are insufficient for global improvement” (Page 11 Line 188)) and explored their impacts on global efficiency.

Moreover, we have also performed a future analysis regarding the structure optimization, from the production shift (primary vs. secondary) as well as the technical shift (low-carbon technology vs. the existing capacity) related to the 1.5 °C target for global warming.

These analyses give rise to several suggestions, i.e. 1) Harnessing emerging low-carbon technologies in emerging steel markets; 2) Radical early-retirement of primary production capacity in China; 3) Moving towards a closed-loop steel cycle in developed nations. These suggestions are presented at the end of the main manuscript starting from Page 15 Line 267.

Comment#2.21

(13) Another issue related to the structure optimization is the redundancy of steelmaking production capacity, such as in China. This comment is an echo of my previous one on the temporal-spatial analysis of the steelmaking industry globally. I suggest the authors add more words on the analysis of excessive capacity and the loading of in-use stocks of steelmaking facilities by targeting the mitigation strategies.

Response:

This is a very good suggestion! We have extended our analysis in the policy suggestion “2) Radical early-retirement of primary production capacity in China” (Page 15 Line 286) to cover this aspect.

Specific changes to manuscript:

Page 15 Line 286:

2) Radical early-retirement of primary production capacity in China. The existing long-lived production facilities can pose a grave threat to the 1.5 °C climate target¹³. As calculated by IEA, the “committed” direct CO₂ emissions from existing steel facilities can reach ~65 Gt from 2019 to 2060, which can nearly exhaust the remaining carbon budget in the 1.5 °C scenario. Thus, a radical early-retirement of primary production capacity is needed. This will be challenging as the global blast furnace fleet is relatively young with an average age of 13 years. Our further MFA (Fig.5a) suggests that China should reduce its primary facilities by ~170 Mt (equivalent to present total primary production capacity in EU and North America) in the next 15 years and further ~500 Mt by 2050. China has shown the ability and willingness for early-retirement of facilities, and around 100-150 Mt of existing facilities have already been retired during 2016-2020⁵⁸. Aside from China, the excessive capacity has been widely considered as one of the main challenges facing the global steel sector at present⁵⁹ (i.e., ~30% (550 Mt) capacity redundancy in 2018), which calls for optimizing the production capacity at a global scale to eliminate their low-efficiency facilities. Other mitigation options in option e) (e.g., BF top gas recycling, fuel-switching, carbon circulation, etc.) should also be applied in the refurbishment and retrofitting of existing production capacities.

Comment#2.22

(14) The readability and resolution of Fig. 2.3A could be improved, and my concern is that the steel product mass allocation method is not self-explained. Additional explanation is necessary to make it clear. How do you come to the percentiles in the section A?

Response:

We have clarified our method, replaced Fig. 2.3A with a high-resolution figure, and rewritten this part as follows, and the percentiles are obtained based on the study by Cullen et al.,² which has been further explained.

Specific changes to manuscript:

² Cullen JM, Allwood JM, Bambach MD. Mapping the global flow of steel: from steelmaking to end-use goods. *Environ Sci Technol* 2012;46:13048–55. <https://doi.org/10.1021/es302433p>.

(b) Steel product mass allocation. This study quantify the steel flows from casting to end-use sectors based on the studies from [2], the relationship of each flow is mapped in Fig.S2.3A (the red figure represents the material flow as noted in Fig. S2.3A). This study assumed all the cold rolling and finishing stage as one process to produce final cold rolling (CR) products, and the production flow from ingot casting (i.e. AA) is allocated to five downstream processes (i.e. 12.1% into cast steel, 6% into section mill, 36.1% into rod/bar mill, 8% into plate mill, and 16.9% into strip mill). For the share of outflow from Rod/Bar Mill and Strip Mill to the CR process, it is based on the historical data from world steel yearbook from 1984 to 2015 and United States case [43] from 1942 to 1984. Meanwhile, the allocation of final steel products (i.e. A-caste steel, B-shapes, C-bars, D-cold-rolling products, E-plates, F-coil and strip, G-cast iron) to four end-use applications (i.e. construction, vehicles, machinery, daily goods) is quantified based on the allocation matrix in [2], which is marked as red in the left box of Fig.S2.3A (i.e. inflow to construction= $0.99*N+0.6*C+0.36*D+0.06*E+0.62*F+0.4*G$).

Note: AA-production of Ingot casting; BB-production of Continuous casting; G - production of iron foundry; A-Cast steel production; B-steel shape production; C-Hot rolling (HR) bar production; D-Cold rolling production; E: HR plate steel production; F-production of HR coil and strip

Fig. S2.3 Steel allocation model (A: Mass allocation from rolling to finishing [2]; B: The historical change of mass share to cold rolling)

Comment#2.23

(15) L223, the citation of (Fig. 4c) is incorrect. It is the same for (Fig. 4b) in L226.

Response:

We have corrected these mentioned citations of figures, and double-checked the whole manuscript to ensure all citations are correct.

Comment#2.24

Detailed comments on the supplementary materials.

(1) SI, L224-226, two symbols, t and t , are not defined. " θ " is not included in the equation.

Response:

This has been revised, and the parameter θ has been deleted.

Specific changes to manuscript:

Supplementary Information 1, Page S16:

where, $f(t, \tau, \sigma)$ is the probability density of the lifetime distribution function; t is the quantification time step; τ is the lifetime of this product sector; σ is the standard deviation of lifetime; T is the end of the studied period; T_0 is the starting time.

Comment#2.25

(2) SI, L306-308, the presentation of parameters is not consistent with the equation S5.

Response:

Thank you for noting this error. This has been corrected.

Specific changes to manuscript:

Supplementary Information 1, Page S26:

S2.2.3 GHG emission calculation

Based on the technology and time differentiated inventory of GHG emissions, the total GHG intensity for each steel production technology was estimated according to Eq. S3.

$$E_i(t) = \sum_x E_{x,i}(t) \times GWP100_x \quad S3$$

Where E_i is the GHG emission intensity of steel production technology i [kg CO₂-eq / kg output] at year t (see Figure S2.16). $E_{x,i}$ is kg emission of GHG x per kg output from steel production technology i at time t . $GWP100_x$ is the global warming potential [kg CO₂-eq / kg GHG _{x} emitted] for GHG x [61] (see Table S2.4). The development in total GHG intensity [kg CO₂-eq / process output] over time for each steel processing process is shown in Figure S2.6.

The total GHG emission per steel production technology per year was estimated as:

$$mGHG_i(t) = E_i(t) \times m_i(t) \quad S4$$

Where $mGHG_i(t)$ is the total emission of CO₂-eq in year t from steel production technology i . $m_i(t)$ is the total output from steel production technology i at time t estimated using the dynamic MFA model. The sum of all CO₂-eq emissions from all steel production processes in year t gave the total emission of CO₂-eq from steel production in year t :

$$mGHG(t) = \sum_i mGHG_i(t)$$

S5

Comment#2.26

(3) Figure S3.8, the color difference between the legends of upper bond value and lower bond value are very minor, it is better to make them different.

Response:

This has been revised. See the updated figure below.

FIG. S3.8 COMPARISON OF ESTIMATED GHG INTENSITIES IN THIS STUDY WITH OTHER GHG INTENSITIES REPORTED IN LITERATURE [48,73-76]

REVIEWERS' COMMENTS

Reviewer #1 (Remarks to the Author):

To the authors -

The paper is radically improved and is nearly ready for publication. There are a bunch of minor English things that need a careful edit, e.g.

- Page 15, line 281, "sitting" not "siting".
- SIDERWIN, not IDERWIN
- Check your sentences for use of "USA", often it should be "the USA"
- page 17, line 337 Australia as "a" centre for iron making (while Gielen et al is great paper, my guess is Australia won't be "the" (only) centre for iron making, or maybe it will ;)),

The end of the paper is much better, but could use a stronger concluding paragraph before the methods section.

Your reviewer

Reviewer #2 (Remarks to the Author):

Comments

I appreciate the authors for their efforts on making a substantial revision of the manuscript and responding my concerns carefully. I think the manuscript has been improved substantially, the response is appropriate, and the manuscript could be considered for acceptance of publication after addressing some minor issues as follows.

1. Abstract is the showcard of the paper. Could you enrich the abstract with much quantitative presentations? The manuscript is heavy with data and quantitative analysis, however, the abstract reads much qualitatively, in particular, the joint supply-and-demand GHG mitigation efforts could be presented in more details.
2. Please check the writing of line 46-48, a verb seems missed.
3. line150-151, 'historical growth in production flow was driven by the emerging nations with poor efficiency performance as shown in their share change between 1995 and 2015'---- Generally, the latecomer of bulk material industries, such like steel manufacturing, will have an opportunity starting with high energy efficiency performance. Could you add more words on the reasons of 'the emerging nations with poor efficiency performance' or add reference?
4. Line 160, 'the GHG intensities of both the primary and secondary production route have stagnated, and remained at ~ 3.0 t CO₂-eq/t steel and ~ 1.0 t CO₂-eq/t steel (Fig. 4c, 4e), respectively'----- 'stagnation is the very important work in this manuscript', I think it is better to zoom in the GHG intensity from 1995-2015 with the year of 1995 as the reference, and then have a high-resolution diagram of the variation of GHG intensity in this period, such will solidify the claim of stagnation. Because the presentation of ' ~ 3.0 t CO₂-eq/t steel and ~ 1.0 t CO₂-eq/t steel' is some ambiguous. Anyhow, you have the data, it better to give a clear scope of the stagnation of GHG intensity in 1995-2015. If possible, a spatial resolution of such stagnation diagram will be useful. There are some other such like presentations as ' ~ 3 -fold' and ' ~ 20 fold', I think it is better to present much accurate with a scope instead of ' \sim '.
5. Line 341, please add a reference to the high carbon taxation (i.e. 100 \$/tCO₂ + 4% per year since 2020)
6. The figure S1.1 looks nice, perhaps it could be added in the main text, the method section, to guide the understanding of the methodology.
7. The layout of Supplementary Information 2 seems disorder.

Response to reviewers' comments on the manuscript:

Efficiency Stagnation in Global Steel Production Urges Joint Supply- and Demand-side Mitigation Efforts

Dear reviewers,

We would like to thank you for your time and large effort in reviewing this manuscript and for providing very valuable and relevant comments on the manuscript to help augmenting its quality.

Your comments are addressed point-by-point in the following (in purple font).

Sincerely,

Morten Ryberg (on behalf of all co-authors)

Response to Reviewer #1

Comment#1.1

Dear authors

The paper is radically improved and is nearly ready for publication. There are a bunch of minor English things that need a careful edit, e.g.

- Page 15, line 281, "sitting" not "siting".
- SIDERWIN, not IDERWIN
- Check your sentences for use of "USA", often it should be "the USA"
- page 17, line 337 Australia as "a" centre for iron making (while Gielen et al is great paper, my guess is Australia won't be "the" (only) centre for iron making, or maybe it will ;))

Response to reviewer:

We are highly appreciative of your encouraging comments and suggestions. We have revised these grammatical errors and have carefully proofread the whole manuscript.

Comment#1.2

The end of the paper is much better, but could use a stronger concluding paragraph before the methods section.

Response:

We have extended the conclusion part to provide a stronger concluding paragraph. It now reads:

Line 360: In conclusion, our study demonstrates that the total carbon reduction not only relies on the implementation of low-carbon technologies but also on their interplay with the changes of production flows at global level. Historical evidence from the steel industry demonstrated those regional process efficiency improvement efforts have not been able to keep up with the growth in production flow, leading to a 17-fold net increase in annual GHG emissions during the studied period. Moreover, we also see that the GHG intensity of steel production at global scale has stagnated in past decades. Thus, it is recommended that key nations and steel producers clarify and develop roadmaps for steel decarbonization, which combine both supply-side and demand-side measures to stop the stagnation and further increase process efficiency as well as reduce the growth in steel demand to be able to sufficiently reduce GHG emissions at global scale. Indeed, it is important that these roadmaps are ambitious enough to meet the steel industry's targets for realizing the 1.5 °C target that was set out in the Paris agreement on climate change.

Response to Reviewer #2

Comment#2.1

I appreciate the authors for their efforts on making a substantial revision of the manuscript and responding my concerns carefully. I think the manuscript has been improved substantially, the response is appropriate, and the manuscript could be considered for acceptance of publication after addressing some minor issues as follows.

Response:

We are highly appreciative of your encouraging comments and suggestions. The response for each point has been given as follows.

Comment#2.2

1. Abstract is the showcard of the paper. Could you enrich the abstract with much quantitative presentations? The manuscript is heavy with data and quantitative analysis, however, the abstract reads much qualitatively, in particular, the joint supply-and-demand GHG mitigation efforts could be presented in more details.

Response:

We understand and appreciate your comment on this. We have made small edits to improve our abstract. However, due to the journal's word limit for the abstract, we are not able to go into further details on the results of the study in the abstract.

The abstract now reads:

Line 26: Steel production is a difficult-to-mitigate sector that challenges climate mitigation commitments. Efforts for future decarbonization can benefit from understanding its progress to date. Here we report greenhouse gas emissions from global steel production over the past century (1900-2015) by combining material flow analysis and life cycle assessment. We find that ~45 Gt steel was produced in this period leading to emissions of ~147 Gt CO₂-eq. The historical improvement in process efficiency (~67%) was offset by a 44-fold increase in annual steel production, resulting in a 17-fold net increase in annual emissions. Despite regional technical improvements, the industry's decarbonization progress stagnated after 1995 mainly due to expansion of emerging production flows with high carbon intensity. This raises concerns on the expected demand expansion in emerging economies, which may jeopardize steel industry's prospects for following 1.5°C emission reduction pathways. This warrants urgent implementations of regionally specific joint supply- and demand-side mitigation measures.

2. Please check the writing of line 46-48, a verb seems missed.

Response:

This has been revised. The sentence now reads:

Line 48: Thus, compared with transportation and energy sectors, the corresponding innovation, progress, and understanding related to the decarbonization of global steel industry is generally lagging behind¹⁴⁻¹⁶.

Comment#2.3

3. line150-151, 'historical growth in production flow was driven by the emerging nations with poor efficiency performance as shown in their share change between 1995 and 2015'----

Generally, the latecomer of bulk material industries, such like steel manufacturing, will have an opportunity starting with high energy efficiency performance. Could you add more words on the reasons of 'the emerging nations with poor efficiency performance' or add reference?

Response:

There are two main reasons: a) Technology factor: The application of new energy-efficient facilities was not always the case in the steel industry of those developing nations, depending on the economic competitiveness in terms of investment and replacement cost of those technologies, and the willingness of technology sharing, etc. b) Flow factor: the growing demand of emerging nations was usually fed with ore-based primary steel production route, which are high carbon intensive that the scrap-based one in developed nations. We have already explained those factors in the "Section: Regional technical efforts are insufficient for global improvement".

We also collected the data of detailed national level comparison in Section S4.3, mainly from the pioneering work from Global Efficiency Intelligence and China Energy Group from Lawrence Berkeley National Laboratory, which can support the statement of "emerging nations with poor efficiency performance".

Accordingly, we made the revision in Line150-151 as follows:

Line 163:by the emerging nations (i.e. China, India, etc.) with poor efficiency performance (described in Section S4.3).....

Comment#2.4

4. Line 160, 'the GHG intensities of both the primary and secondary production route have stagnated, and remained at ~3.0 t CO₂-eq/t steel and ~1.0 t CO₂-eq/t steel (Fig. 4c, 4e), respectively'----- 'stagnation is the very important work in this manuscript', I think it is better to zoom in the GHG intensity from 1995-2015 with the year of 1995 as the reference, and then have a high-resolution diagram of the variation of GHG intensity in this period, such will solidify the claim of stagnation. Because the presentation of '~3.0 t CO₂-eq/t steel and ~1.0 t CO₂-eq/t steel' is some ambiguous. Anyhow, you have the data, it better to give a clear scope of the stagnation of GHG intensity in 1995-2015. If possible, a spatial resolution of such stagnation diagram will be useful. There are some other such like presentations as '~3-fold' and '~20 fold', I think it is better to present much accurate with a scope instead of '~'.

Response:

Thanks for your suggestion. We have tried to merge the high-resolution figures of the variation of GHG intensity since 1990 into Fig. 4a, c, e. and this will confuse the overall presentation. Thus, we decided to move the following figure into Fig. S3.9 of section S3.4 in Supplementary Information 1 to highlight the stagnation that occurs during this period.

As for the spatial resolution of such stagnation, the information is not available due to the lack of continuous and comparable records of energy (process) efficiency in different nations (the best available records were summarized in section S4.3).

Finally, the numerical presentations such as “~3-fold” have been improved and made more accurate.

Comment#2.5

5. Line 341, please add a reference to the high carbon taxation (i.e. 100 \$/tCO₂ + 4% per year since 2020)

Response:

This has been added. This information is from *Ruijven, B. J. Van et al. Long-term model-based projections of energy use and CO₂ emissions from the global steel and cement industries. Resour. Conserv. Recycl. 112, 15–36 (2016). <https://doi.org/10.1016/j.resconrec.2016.04.016>*

Comment#2.6

6. The figure S1.1 looks nice, perhaps it could be added in the main text, the method section, to guide the understanding of the methodology.

Response:

Thank you for the comment on this figure and the good suggestion. The method part does not usually include a figure, and we may keep this in Supplementary Information 1 to guide readers for a good understanding of the methodology detailed in the Supplementary Information 1.

Comment#2.7

7. The layout of Supplementary Information 2 seems disorder.

Response:

Thank you for pointing this out. We have improved the layout and organization of this file, which is now referred to as Supplementary Data 1.